# Distinct mechanisms of microRNA sorting into cancer cell-derived extracellular vesicle subtypes

Morayma M Temoche-Diaz[1], Matthew J Shurtleff[1,2], Ryan M Nottingham[3], Jun Yao[3], Raj P Fadadu[2], Alan M Lambowitz[3], Randy Schekman[2]*

[1]Department of Plant and Microbial Biology, University of California, Berkeley, Berkeley, United States; [2]Department of Molecular and Cellular Biology, Howard Hughes Medical Institute, University of California, Berkeley, Berkeley, United States; [3]Department of Molecular Biosciences, Institute for Cellular and Molecular Biology, University of Texas, Austin, United States

**Abstract** Extracellular vesicles (EVs) encompass a variety of vesicles secreted into the extracellular space. EVs have been implicated in promoting tumor metastasis, but the molecular composition of tumor-derived EV sub-types and the mechanisms by which molecules are sorted into EVs remain mostly unknown. We report the separation of two small EV sub-populations from a metastatic breast cancer cell line, with biochemical features consistent with different sub-cellular origins. These EV sub-types use different mechanisms of miRNA sorting (selective and non-selective), suggesting that sorting occurs via fundamentally distinct processes, possibly dependent on EV origin. Using biochemical and genetic tools, we identified the Lupus La protein as mediating sorting of selectively packaged miRNAs. We found that two motifs embedded in miR-122 are responsible for high-affinity binding to Lupus La and sorting into vesicles formed in a cell-free reaction. Thus, tumor cells can simultaneously deploy multiple EV species using distinct sorting mechanisms that may enable diverse functions in normal and cancer biology.
DOI: https://doi.org/10.7554/eLife.47544.001

*For correspondence:
schekman@berkeley.edu

## Introduction

Extracellular vesicles (EVs) are membranous compartments that consist of a lipid bilayer with a unique set of transmembrane proteins enclosing soluble contents that include nucleic acids and proteins (*Colombo et al., 2014*). EVs are found in biofluids as well as in conditioned media of cultured cells (*Pisitkun et al., 2004*; *Caby et al., 2005*; *Admyre et al., 2007*; *Vella et al., 2007*). Cells release an array of EV sub-populations, which can be broadly classified into two categories: exosomes and shedding vesicles, distinguished by the cell membrane of origin. Exosomes are 30–150 nm vesicles that originate in the endocytic pathway. Their secretion to the extracellular space occurs upon multi-vesicular body (MVB) fusion with the plasma membrane resulting in the release of intraluminal vesicles (ILVs) to the extracellular space (*Harding et al., 1983*; *Pan et al., 1985*). Shedding vesicles, or microvesicles, refer to a more heterogeneous group of EVs, with sizes ranging from 30 to 1,000 nm, which originate by budding directly from the plasma membrane (*Cocucci et al., 2009*). Shedding vesicles and exosomes are also termed as large and small EVs respectively. Their isolation is generally achieved by differential ultracentrifugation. Evidence, however, that multiple EV species are being co-isolated as small EVs (sEVs) has arisen, proving that the sEV fraction represents a crude mix of different EV species. Studies further purifying different sEVs sub-populations have shown that they carry different protein signatures (*Kowal et al., 2016*). Moreover, distinct sEV sub-populations have been implicated in mediating various physiological responses (*Willms et al., 2016*). Despite the

current knowledge of sEV heterogeneity, most current methods used to isolate sEVs do not distinguish between sEV sub-populations, nor do they separate sEVs from protein and RNA-protein complexes (RNPs) that are not EV encapsulated (*Shurtleff et al., 2018*). New methods to facilitate the purification of sEV sub-populations are needed in order to evaluate the physiological role and mechanism of macromolecular sorting into each species.

The lipid membrane of EVs confers protection to its soluble contents against extracellular hydrolases, thus increasing their stability in the extracellular space. The EV soluble content consists of a variety of proteins as well as small RNA molecules (*Raposo and Stoorvogel, 2013*). Among the EV resident transcripts, miRNAs have garnered special attention. MiRNAs are ~22 nucleotide transcripts that modulate gene expression at the post-transcriptional level (*Bartel, 2004*). Although the functional importance of EV-miRNAs, specially sEV-miRNAs, as signaling molecules has received some support, including in immunologic response and metastatic tumor cell growth (*Mittelbrunn et al., 2011*; *Montecalvo et al., 2012*; *Pegtel et al., 2010*; *Fong et al., 2015*; *Dickman et al., 2017*; *Zhou et al., 2014*; *Tominaga et al., 2015*; *Hsu et al., 2017*), the molecular mechanisms and regulation of sorting miRNAs into sEVs remain poorly understood. Both, non-selective (*Tosar et al., 2015*) and selective (*Shurtleff et al., 2016*; *Santangelo et al., 2016*; *Villarroya-Beltri et al., 2013*) mechanisms of miRNA sorting into EVs have been described. One study using two breast epithelial cell lines found that non-selective miRNA sorting occurs in large EVs, referred to as 'large-exosomes', whereas a selective mechanism of sorting occurs in sEVs (*Tosar et al., 2015*). The sEV fraction however was not further purified into sEV sub-populations. Therefore the conclusion that selective sorting occurs in the sEV fraction could have been the result of only one or few sEV sub-populations skewing the results.

Cancer cells produce more EVs than their non-transformed counterparts (*Szczepanski et al., 2011*; *Rodríguez et al., 2014*). Thus, EVs have been the focus of attention in the context of cancer biology. Several studies have implicated EVs in cancer metastasis (*Peinado et al., 2017*). EVs, especially sEVs, have been proposed to increase the metastatic potential of cancer cells and to prepare the pre-metastatic niche prior to tumor cell arrival. Importantly, some studies suggest these effects are due to exosomal-mediated miRNA transfer (*Fong et al., 2015*; *Zhou et al., 2014*; *Tominaga et al., 2015*; *Rana et al., 2013*; *Hsu et al., 2017*). However, due to the heterogeneity of sEVs and the procedures used to isolate them, it is unclear which vesicle subtypes may mediate these processes.

To gain knowledge of the molecular mechanisms behind miRNA sorting into distinct sEV sub-populations, we used as a model the highly invasive breast cancer cell line MDA-MB-231. We used buoyant density gradient centrifugation to resolve two distinct sEV sub-populations and found they contain different subsets and levels of enrichment of various miRNAs. With the aid of a cell-free reaction that recapitulates the sorting of miRNAs into vesicles formed in vitro (*Shurtleff et al., 2016*), we found that the Lupus La protein (La protein) is required for the selective sorting of miRNAs found in the high buoyant density sEV species, a conclusion that was confirmed in cells repressed for the expression of the La protein. At least two La-dependent miRNAs were previously reported to be selectively sorted into sEVs to promote cancer metastasis (*Fong et al., 2015*; *Dickman et al., 2017*). Further, using the cell-free reaction, we identified two La-binding motifs that are required for the selective sorting of miR-122 in vitro. Overall, our use of biochemical fractionation and in vitro reconstitution enabled us to identify La as another one of the RNA-binding proteins required for the selective packaging of some miRNAs into sEVs.

## Results

### Two biochemically distinct small extracellular vesicle sub-populations are released by MDA-MB-231 cells

To examine the miRNA content of sEVs released by the metastatic breast cancer cell line MDA-MB-231, we first sought to separate sEVs from contaminating RNA and protein species that co-sediment during common EV isolation procedures (*Shurtleff et al., 2018*). To achieve this goal, we developed an EV purification strategy involving differential ultracentrifugation followed by buoyant density flotation on a linear iodixanol gradient (*Figure 1a* and *Figure 1—figure supplement 1*). This approach allowed the resolution of two distinct sEV species, termed the vesicular

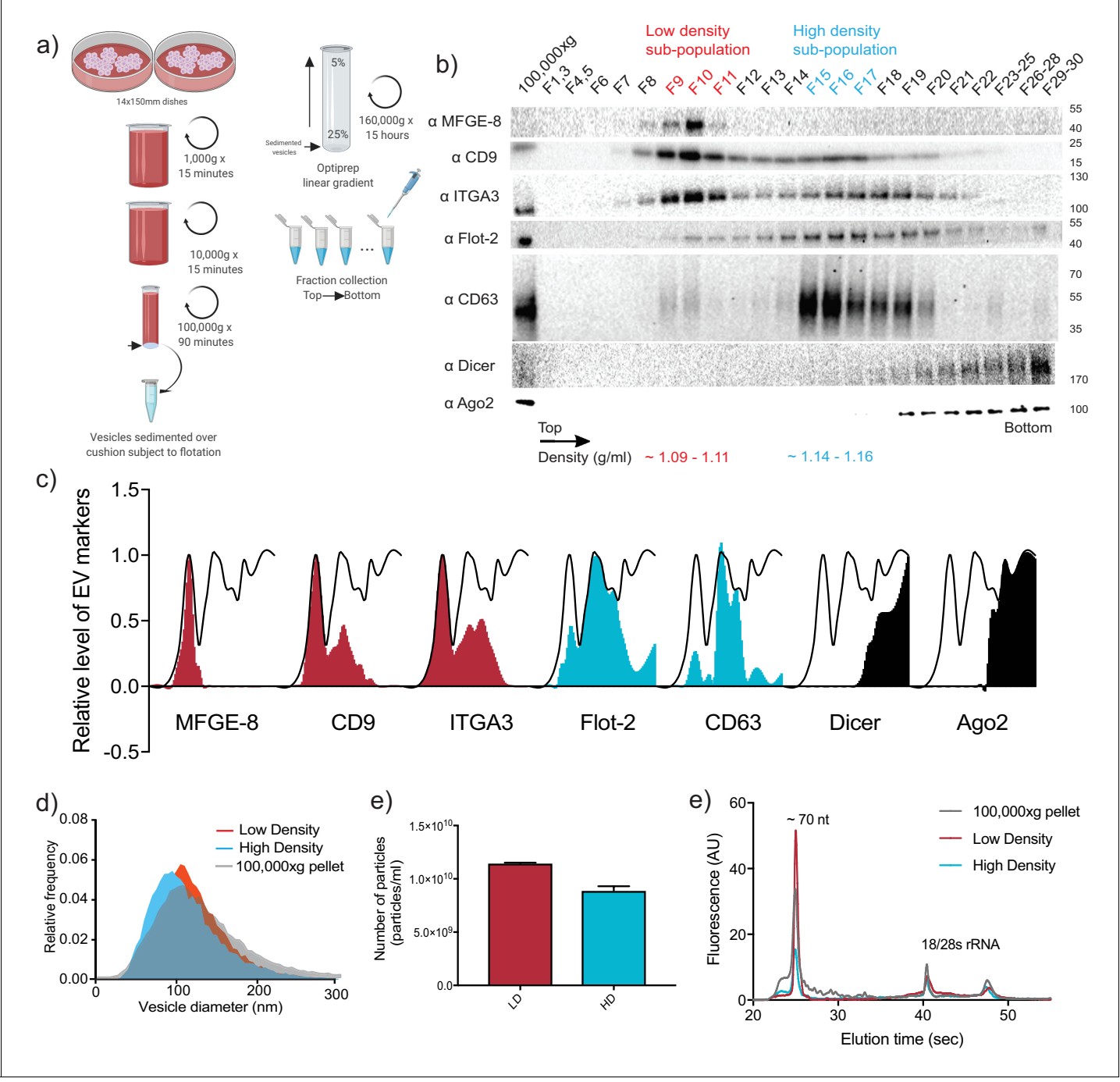

**Figure 1.** Two biochemically distinct sEV sub-populations are released by MDA-MB-231 cells. (**a**) Schematic showing the two-step purification methodology. Differential ultracentrifugation was followed by buoyant density flotation in a linear iodixanol gradient (***Figure 1—figure supplement 1***). (**b**) Immunoblot across the iodixanol gradient for classical sEV markers as well as other non-vesicular RNP components. The two discrete sub-populations are indicated. CD63, a glycosylated protein, migrates heterogeneously. (**c**) Quantification of individual proteins as in 1b. The black line delineates the maximum signal across the gradient showing three distinctive areas. The first, second and third peak represent the vLD, vHD and non-vesicular RNP fractions. (**d**) Nanoparticle tracking analysis showing the size distribution of the vHD and vLD sub-populations. The high-speed pellet is also shown. (**d**) Quantification of the particle number per sEV sub-population using Nanoparticle tracking analysis. Data plotted are from two independent experiments; error bars represent standard deviation from independent samples. (**e**) Bioanalyzer analysis of the vHD and vLD RNA. The high-speed pellet RNA is also shown.

DOI: https://doi.org/10.7554/eLife.47544.002

The following source data and figure supplements are available for figure 1:

*Figure 1 continued on next page*

*Figure 1 continued*

**Figure supplement 1.** Linearity of iodixanol density gradient.
DOI: https://doi.org/10.7554/eLife.47544.003
**Figure supplement 1—source data 1.** Refraction index and calculated densities of the fractions across the gradient (31 fractions in total).
DOI: https://doi.org/10.7554/eLife.47544.004

low density (vLD) and vesicular high density (vHD) sub-populations (*Figure 1b*). Both sEV sub-populations fractionated away from non-vesicular RNPs, as shown by the presence of Ago2 and Dicer in the denser areas of the gradient (*Figure 1b*). This result is in accordance with previous studies finding the presence of Ago2 in the crude 100,000 g pellet, but not associated with vesicular fractions when more stringent methods of purification were applied (*Shurtleff et al., 2016*; *Van Deun et al., 2014*; *Jeppesen et al., 2019*). Quantification of each protein marker along the gradient showed that classical EV reporters exhibited a differential distribution to one or the other of the sEV sub-populations, whereas Ago2 and Dicer did not float and remained in the denser fractions (*Figure 1c*).

A recent study reported the buoyant density separation of vesicular and non-vesicular particulate material in the high-speed pellet fraction of cell culture-conditioned medium (*Jeppesen et al., 2019*). Low buoyant density vesicles and non-vesicular high density fractions were resolved on a 12–36% iodixanol gradient. Our results document that Ago2 and Dicer remain in the higher density (bottom) of the gradient (*Figure 1b* and *Figure 4—figure supplement 1*), separated from membrane protein markers of EVs. The use of a slightly modified iodixanol gradient (5–25%), and the collection of 30 fractions (*Figure 1b*) permitted a higher level of membrane resolution including the separation of two distinct buoyant density vesicle species, termed vLD and vHD.

Nanoparticle tracking analysis of the two sEV sub-populations demonstrated that vesicle sizes largely overlapped, with vHD vesicles (114.1 nm mean diameter) being slightly smaller on average than vLD vesicles (121.8 nm mean diameter) (*Figure 1d*). Quantification of the total particle number of both sEV sub-populations showed that there were approximately 30% more vLD vesicles released into the medium in comparison to the vHD counterpart (*Figure 1e*). The vHD sub-population had densities ranging from 1.14 to 1.16 g/ml as was reported for exosomes (*Colombo et al., 2014*). Furthermore, CD63, a tetraspanin enriched in MVBs and an exosomal marker, was highly enriched in the vHD fraction (*Figure 1b and c*) as were other endosome-associated proteins identified by mass spectrometry (*Figure 2—source data 1*). The vLD sub-population had a density ranging from 1.09 to 1.11 g/ml, had greatly reduced CD63 signal (*Figure 1b*) and lacked endosome-associated proteins as judged by mass spectrometry (*Figure 2—source data 1*).

Gene ontology analysis for sub-cellular localization of the proteins detected in the vHD sub-population by mass spectrometry clustered the terms exosomes and endosomes together (*Figure 2a*). The vLD-detected proteins showed exosomal marker proteins, but endosomal markers were not found at statistically significant levels (*Figure 2b*). We then analyzed the sub-cellular localization of two tetraspanins enriched in each sEV sub-population by immunofluorescence microscopy: CD9 and CD63, enriched in vLD and vHD respectively (*Figure 1b*). The two analyzed tetraspanins showed a distinctive sub-cellular localization, with CD9 mostly localizing in the plasma membrane (*Figure 2c*) and CD63 displaying an intracellular localization (*Figure 2d*). This result is in accordance with a recent study showing that CD9 has a mainly a plasma membrane sub-cellular localization, whereas CD63 localizes to endosomes (*Gould et al., 2019*). Thus, we hypothesize that the vHD and vLD vesicles have different sub-cellular origins.

The Rab27a protein is involved in the fusion of MVBs to the plasma membrane, regulating in this way exosomal secretion (*Ostrowski et al., 2010*). We created a Rab27a knock out cell line using CRISPR-Cas9 technology (*Figure 2e*). Extracellular vesicles were isolated from conditioned medium of WT and Rab27a knock out cells and analyzed by immunoblot for the levels of two tetraspanins defining the two sEV sub-populations: CD63 as a protein marker defining the vHD sub-population and CD9 defining the vLD sub-population (*Figure 1b*). sEVs derived from the Rab27a KO background showed a dramatic decrease of CD63 signal, while secreting seemingly similar levels of CD9-positive vesicles (*Figure 2f*) in comparison to the WT levels. Our results are in accordance with a previous study showing that the secretion of CD63-positive, but not of CD9-positive vesicles, is Rab27a dependent (*Bobrie et al., 2012*). However, despite the dramatic reduction of secretion of CD63

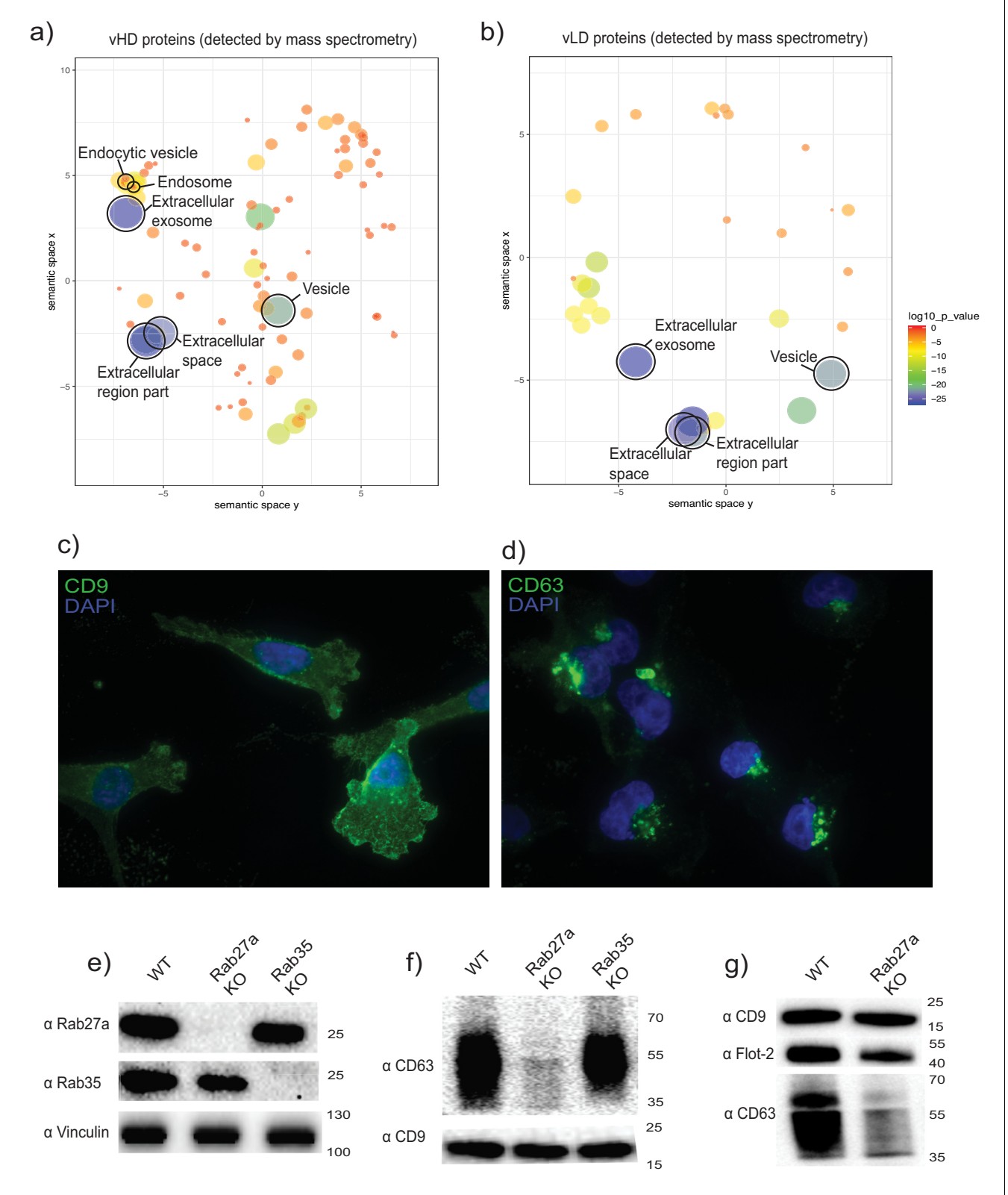

**Figure 2.** The two biochemically distinct sEV sub-populations co-fractionate with membranes of different sub-cellular origin. (a, b) Gene ontology analysis for sub-cellular localization of proteins that coincide with vHD and vLD proteins detected by mass spectrometry. The gene ontology analysis platform compares the list of proteins given per group to the human genome frequency. The detection of enrichment for a certain organelle/localization is represented as circles, with the size of circles correlating to log10 p-value. (c, d) Immunofluorescence for two tetraspanins enriched in

*Figure 2 continued on next page*

*Figure 2 continued*

each sEV sub-population. CD9 and CD63 enriched in vLD and vHD respectively. (**e**) Analysis of WT, CRISPR-Cas9 Rab27a KO and CRISPR-Cas9 Rab35 KO by immunoblot. Rab27a, Rab35 and vinculin are shown. (**f**) Immunoblots for CD63 and CD9 in sEVs secreted by WT, Rab27a KO cells and Rab35 KO cells. The amounts of loaded sEVs were normalized by total cell number. (**g**) Immunoblots for CD63, CD9 and flotillin-2 in sEVs secreted by WT and Rab27a KO cells. The amounts of loaded sEVs were normalized by total cell number.

DOI: https://doi.org/10.7554/eLife.47544.005

The following source data is available for figure 2:

**Source data 1.** List of proteins detected by mass spectrometry in both EV sub-populations.

DOI: https://doi.org/10.7554/eLife.47544.006

positive vesicles under the Rab27a KO background, there was only a subtle reduction for the vHD enriched marker flotillin-2 (*Figure 2g*). This suggests that the vHD vesicle fraction may comprise at least two different types of sEV sub-populations, with at least one of them having its origin in the endocytic pathway. This possibility is consistent with our previous demonstration that sEVs secreted by HEK293T cells contain a population of vesicles that bind to immobilized CD63 antibody and a fraction containing flotillin-2 that does not (*Shurtleff et al., 2017*).

The Rab35 protein has also been implicated in mediating exosomal secretion in oligodendrocytes (*Hsu et al., 2010*). Thus, we created a Rab35 knock out cell line (*Figure 2c*). When sEVs derived from the Rab35 KO background were probed for CD63 and CD9 signals, no apparent difference was observed in comparison to EVs derived from WT cells (*Figure 2e*). Thus, Rab27a but not Rab35, is involved in the secretion of CD63-positive vesicles derived from MDA-MB-231 cells. Collectively, our data are most consistent with vHD vesicles having at least one type of vesicle originating in the endocytic pathway, whereas vLD vesicles may bud directly from the plasma membrane.

We next probed the small RNA content of the separated vesicle fractions. An initial qualitative assessment of RNA profiles using a Bioanalyzer showed that both vLD and vHD sub-populations share a predominant peak at ~70 nucleotides (*Figure 1e*), which later proved to correspond to tRNAs (*Figure 3—figure supplement 2d*). However, there was an apparent enrichment of transcripts smaller than 70 nucleotides in the vHD sub-population (*Figure 1e*) that we hypothesized represented miRNAs. As such, we considered the possibility that the distinct sEV sub-populations might differ in their miRNA content.

## Distinct molecular mechanisms of miRNA sorting govern the discrete small extracellular vesicle sub-populations

To further analyze the miRNA content of the vHD and vLD vesicles, we used two orthogonal approaches: targeted miRNA profiling and an unbiased sequencing approach. Using the commercially available Firefly particle technology, Discovery Panel (Abcam), we compared the abundance of 408 miRNAs in the vLD and vHD sEV sub-populations. Out of the ~400 tested miRNAs, 312 showed a detectable signal in the vHD sub-population and 217 were detected in the vLD counterpart. We found 21 miRNAs present in the vHD vesicles that were enriched at least 10-fold in comparison to their intracellular levels (*Figure 3a*). In contrast, only two miRNAs detected in the vLD sub-population showed over a 10-fold change relative to cell lysate (*Figure 3b*). Both of the apparent vLD-enriched miRNAs were also present as highly enriched in the vHD sub-population (*Figure 3a*). Their enrichment in the vLD sub-population was much lower than in the vHD sub-population (*Figure 3—figure supplement 2*), suggesting their detection in the vLD subpopulation may have been the result of small amounts of vHD vesicles co-fractionating with the vLD vesicles during flotation. Validation by RT-qPCR suggested that these enriched miRNAs were highly specific to the vHD sub-population (*Figure 3c*), confirming our initial suspicion.

As an orthogonal and unbiased approach to profile EV miRNA, we used thermostable group II intron reverse transcriptase sequencing (TGIRT-seq). The TGIRT enzyme is a thermostable, highly processive reverse transcriptase with a proficient template-switching activity used for RNA-seq adapter addition (*Mohr et al., 2013*). This enzyme's properties allowed us to generate comprehensive cDNA libraries that included highly structured and modified transcripts. Also due to its properties, TGIRT allows the generation of cDNA libraries from low input material (*Qin et al., 2016*;

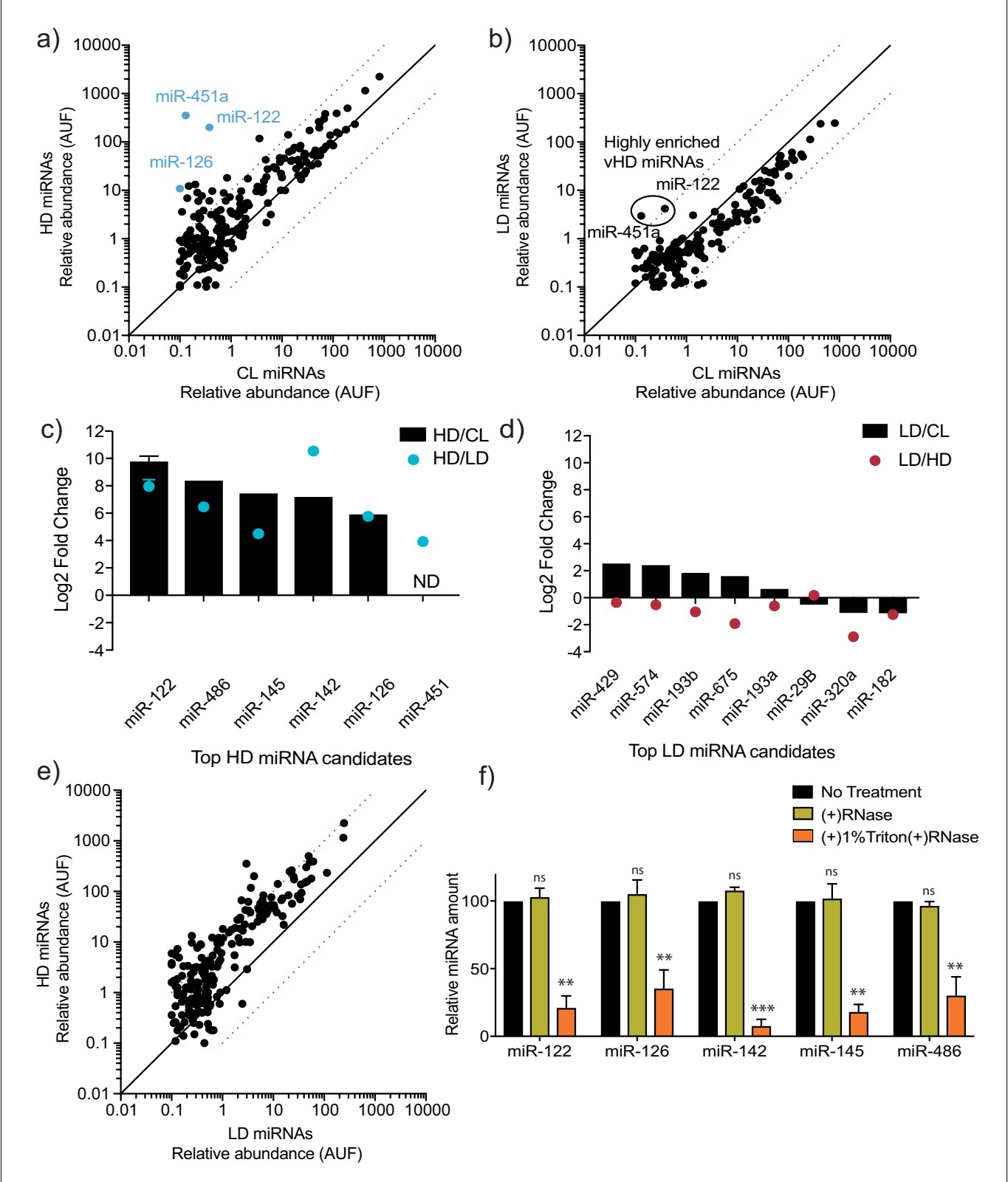

**Figure 3.** MicroRNA profiling of high buoyant density and low buoyant density sEV sub-populations. (a, b) Scatterplots showing the relative abundance (as arbitrary units of fluorescence (AUF)) for miRNAs detected in vHD and vLD sub-populations relative to cellular levels, as detected by Firefly profiling and normalized per ng of total RNA. In blue, vHD miRNAs that were selected for further validation by RT-qPCR. Circled miRNAs in vLD represent those also found to be enriched in the vHD-subpopulation. (c, d) Log2 fold change for top vHD and vLD candidate miRNAs in vHD or vLD relative to cells

*Figure 3 continued on next page*

*Figure 3 continued*

(vHD/CL or vLD/CL, respectively) and vHD relative to vLD or vice versa (vHD/vLD or vLD/vHD, respectively). MiRNA species were quantified by RT-qPCR and normalized per ng of total RNA. Data plotted for miR-122 are from three independent experiments (biological replicates), each independent experiment with triplicate qPCR reactions; error bars represent standard deviation from independent samples. (e) Scatterplots showing relative abundance of miRNAs (AUF) detected in vHD and vLD sub-populations normalized per ng of total RNA, as detected by Firefly profiling. (f) RNase protection of highly enriched vHD miRNAs quantified by qPCR. Isolated sEVs were treated with or without RNase If and or Triton X-100. Data plotted are from two independent experiments, each independent experiment with triplicate qPCR reactions; error bars represent standard deviation from independent samples. Statistical significance was performed using Student's t-test (**p<0.01, ***p<0.001, ns = not significant). CL: cellular lysate. Dashed red lines in a, b and e, represent 10-fold differences.

DOI: https://doi.org/10.7554/eLife.47544.007

The following figure supplements are available for figure 3:

**Figure supplement 1.** MiR-122 and miR-451 are highly enriched in the vHD sub-population.

DOI: https://doi.org/10.7554/eLife.47544.008

**Figure supplement 2.** TGIRT-sequencing of high buoyant density and low buoyant density sEV sub-populations.

DOI: https://doi.org/10.7554/eLife.47544.009

**Figure supplement 3.** MiR-122 accumulates in conditioned media (CM) over time.

DOI: https://doi.org/10.7554/eLife.47544.010

**Figure supplement 4.** MiR-122 co-fractionates with the vHD sub-population.

DOI: https://doi.org/10.7554/eLife.47544.011

*Shurtleff et al., 2017*). The data obtained from TGIRT-seq was normalized to total number of small non-coding RNA reads per sample. As previously described for EVs using TGIRT-seq (*Shurtleff et al., 2017*), we found that miRNAs represented a relatively low proportion of total reads (3.4% and 1% of the mapped reads for vHD and vLD, respectively), resulting in detection of fewer miRNAs by TGIRT-seq than by Firefly profiling. Nevertheless, TGIRT-seq recapitulated the general trends observed by Firefly: multiple miRNAs enriched by at least 10-fold in the vHD sample compared to cells (*Figure 3—figure supplement 2a*) and fewer enriched miRNAs in the vLD sample compared to cells (*Figure 3—figure supplement 2b*). Out of the four apparent vLD enriched miRNA detected by TGIRT-seq, three were also detected as highly enriched in the vHD sub-population, suggesting again that their presence was a result of carryover material overlapping the buoyant density gradient fractions. The total number of uniquely detected vHD miRNAs by TGIRT-seq (detectable in the vHD sample, but absent in the intracellular lysate) was 21. Comparatively, only five miRNAs were uniquely found in the vLD sub-population (*Figure 3—figure supplement 2c*).

In agreement with our previous findings (*Shurtleff et al., 2017*), we found that the most abundant small non-coding transcripts in EVs are tRNAs (representing 88.6% of vHD and 96.3% of vLD transcripts – *Figure 3—figure supplement 2d*). We found that most of the tRNA reads in both sEV sub-populations start at approximately position 16 (*Figure 3—figure supplement 2e*), which may represent a reverse transcription stop at the same unidentified EV-enriched D-loop modification described previously (*Shurtleff et al., 2017*). Thus, both sEV sub-populations share a modified tRNA version as their most abundant small non-coding transcript.

We next validated the top sEV-enriched/unique miRNAs per vesicle sub-population using RT-qPCR. Relative levels for each sEV sub-population were compared to intracellular levels. We found that all of our top vHD candidates were validated by this method. As expected, the tested vHD miRNAs were highly enriched in the vHD sEV sample relative to the intracellular levels. The vHD/intracellular lysate ratio ranged from ~50-fold to ~1,000-fold (*Figure 3c*). In contrast, none of our top vLD miRNA candidates were validated as being highly enriched in comparison to their intracellular levels. The vLD/intracellular lysate ratio ranged from ~0.5-fold to ~5-fold (*Figure 3d*). We then compared relative levels for each EV sub-population to each other in order to test for miRNA vesicle specificity. vHD-enriched miRNAs were highly depleted in the vLD vesicles, as seen by >10-fold increase in their vHD/vLD ratio (*Figure 3c*). In contrast, the tested vLD miRNAs had roughly the same abundance or were depleted in comparison to their relative levels in vHD vesicles (*Figure 3d*). This proved to be true not only for the few miRNAs tested by RT-qPCR, but was also a general trend for multiple miRNAs detected by Firefly profiling, which demonstrated that many miRNAs were more abundant in the vHD vesicles than in their vLD counterpart (*Figure 3e*). All of this together

suggested that a selective mechanism of miRNA sorting occurs in the vHD sub-population with much less or no miRNA selective sorting in the vLD pool.

Highly enriched (>50-fold change vHD/cellular lysate) and specific (>20-fold change vHD/vLD) vHD miRNAs were then tested to confirm that they were bona fide EV residents, and not simply EV-associated. The high-speed pellet fraction was exposed to ribonuclease in the absence or presence of non-ionic detergent. All of the highly enriched, specific vHD miRNAs tested proved to be EV encapsulated: they were resistant to RNase in the absence but not the presence of detergent (*Figure 3f*).

These results suggest that a selective miRNA sorting mechanism into sEVs occurs in the biogenesis of the vHD sub-population but less so or not at all in the vLD sub-population. Thus, we identified two distinct sEV sub-populations that are released by MDA-MB-231 cells that utilize fundamentally different miRNA sorting mechanisms (selective vs non-selective).

## Selective miR-122 sorting into vesicles is recapitulated in vitro

We selected miR-122 for further biochemical analysis as it has been shown that circulating miR-122 can serve as a prognostic biomarker for metastasis in patients with breast cancer (*Wu et al., 2012*) and very specifically that MDA-MB-231 cell-derived exosomal miR-122 may promote metastasis by reprogramming glucose metabolism in the premetastatic niche (*Fong et al., 2015*). Because miR-122 is present in the fetal bovine serum (FBS) used to culture cells (*Wei et al., 2016*), we employed clarified FBS depleted of sedimentable particles to confirm that miR-122 was released by MDA-MB-231 cells. We observed an accumulation of miR-122 over time in conditioned media exposed to MDA-MB-231 cells, whereas the level of miR-122 slightly decreased over time when conditioned media was incubated in the absence of cells (*Figure 3—figure supplement 3*). This suggested that newly synthesized MDA-MB-231 cell-derived miR-122 was secreted into the extracellular space. Moreover, when the levels of miR-122 were tested across the iodixanol linear gradient, we observed co-fractionation of miR-122 with the vHD fraction, substantially separated from RNP particles at the bottom of the gradient (*Figure 3—figure supplement 4a,b and c*). Thus confirming our previous observation that miR-122 is a vHD specific transcript (*Figure 3c*).

To explore the mechanism(s) by which miR-122 is selectively sorted into the MDA-MB-231 vHD sEV sub-population, we first sought to recapitulate miR-122 packaging in vitro. An assay for the cell-free packaging of a miRNA into putative exosomal vesicles was previously developed in our laboratory (*Shurtleff et al., 2016*). The reaction includes sedimented membranes and clarified cytosol, obtained from mechanically ruptured HEK293T cells, along with synthetic miRNA and ATP. After an incubation of 20 min at 30°C, RNA incorporated into vesicles was monitored by protection against degradation by exogenous RNase. Protected RNA was then extracted and quantified by RT-qPCR. The outcome of the reaction is represented as the percentage of initial input miRNA that has become protected over the incubation period (*Figure 4a*). Using a modified version of our previously published reaction, we found that maximum protection of miR-122 in lysates of MDA-MB-231 cells required membranes, cytosol and incubation at a physiological temperature (*Figure 4b*), as was previously observed for miR-223 in lysates of HEK293T cells (*Shurtleff et al., 2016*). Therefore, miR-122 packaging into vesicles in lysates of MDA-MB-231 cells may be recapitulated in vitro.

## Identifying in vitro miR-122 RNA-binding protein partners

Having detected miR-122 packaging in vitro, we next sought to utilize this approach to study the mechanism of miR-122 sorting into MDA-MB-231 sEVs. We first tested if miR-122 packaging was YBX1-dependent. YBX1 is a RNA-binding protein (RBP) previously found to be required for miR-223 packaging into vesicles in vitro and into sEVs secreted by HEK293T cells (*Shurtleff et al., 2016*). Using membranes and cytosol from WT and *ybx1* null HEK293T cells, we repeated the demonstration that miR-223 packaging was YBX1-dependent, but found that miR-122 was packaged nearly normally in lysates devoid of YBX1 protein (*Figure 4c*).

Several RBPs have been implicated in miRNA sorting into sEVs from different cell types (*Shurtleff et al., 2016*; *Mukherjee et al., 2016*; *Santangelo et al., 2016*; *Villarroya-Beltri et al., 2013*). As such, in order to study the RBP(s) that might mediate miR-122 packaging in MDA-MB-231 cells, we performed an in vitro packaging reaction employing a 3'-biotinylated form of miR-122 to allow the capture of the miRNA and any bound proteins. Briefly, following in vitro packaging,

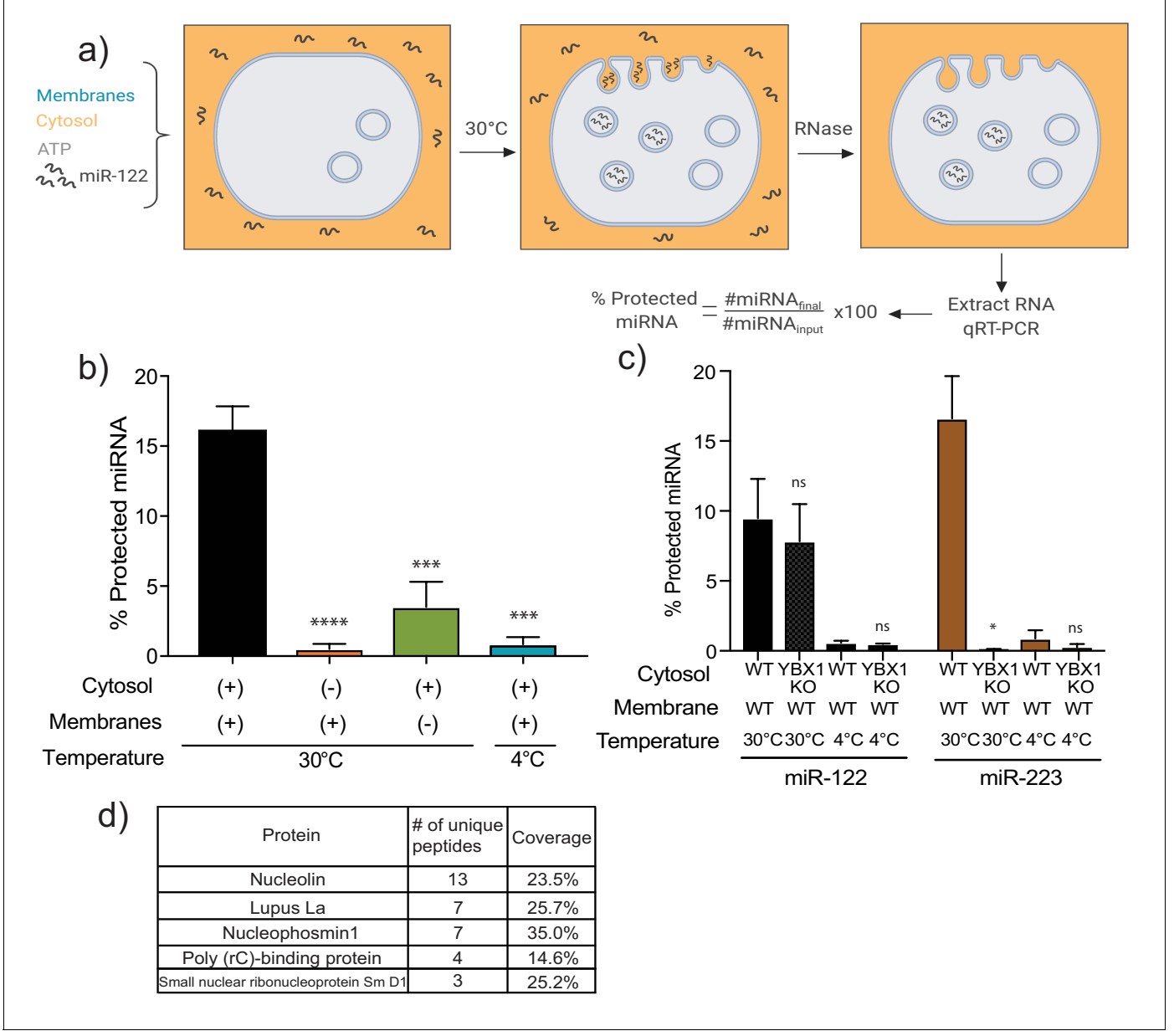

**Figure 4.** MiR-122 packaging is recapitulated in a cell-free reaction. (a) Schematic depicting the in vitro packaging reaction. Image reproduced from the original manuscript developing the cell-free reconstitution assay (*Shurtleff et al., 2016*). (b) Quantification of the in vitro packaging reaction of miR-122. Reactions with or without membranes (15,000xg pellet), cytosol (150,000xg supernatant) prepared from MDA-MB-231 cells and incubated at 30°C or 4°C are shown. Data plotted are from three independent experiments, each independent experiment with triplicate qPCR reactions; error bars represent standard deviation from independent samples. (c) Quantification of the in vitro packaging of miR-122 and miR-223. Cytosols from WT or YBX1 KO HEK293T cells were used. Incubations at 30°C or 4°C are shown. Data plotted are from two independent experiments, each independent experiment with triplicate qPCR reactions; error bars represent standard deviation from independent samples. (d) List of RNA binding proteins pulled down with biotinylated miR-122 after a modified in vitro packaging reaction was performed (see Materials and methods). Statistical significance was performed using Student's t-test (*p<0.05, ***p<0.001, ****p<0.0001, ns = not significant).

DOI: https://doi.org/10.7554/eLife.47544.012

The following figure supplements are available for figure 4:

**Figure supplement 1.** Ago2 and Dicer are not sEV-associated in MB-MDA-231 cells.

DOI: https://doi.org/10.7554/eLife.47544.013

**Figure supplement 2.** CRISPRi efficiently depletes the RNA binding protein candidates.

DOI: https://doi.org/10.7554/eLife.47544.014

reactions were treated with RNase, the RNase activity was quenched and the membranes solubilized with Triton X-100. Once the luminal content was released, biotinylated miR-122, along with its protein interactors, was captured with streptavidin beads. Proteins were eluted with Laemmli buffer, extracted from a SDS-polyacrylamide gel and the eluted fraction used for mass spectrometry. The proteins detected by mass spectrometry were curated for RBPs, except that any ribosomal proteins were excluded (*Figure 4d*). We decided to focus on the top three candidates, nucleolin (NCL), Lupus La (La) and nucleophosmin (NPM1). These three RBPs have been reported to be present in crude high-speed pellet preparations from conditioned media from different carcinoma cell lines (*Liang et al., 2013*; *Demory Beckler et al., 2013*), including from our model cell line MDA-MD-231 (*Skottvoll et al., 2018*).

Notably, Ago2 was not detected bound to miR-122 in our mass spectrometry results. This was not simply an artifact of our in vitro system, as Ago2 was also undetectable in immunoblots of the buoyant density fractionated sEV membranes (*Figure 1b* and *Figure 4—figure supplement 1*). Both, Ago2 and Dicer were present in the high-speed pellet, but not as buoyant species, suggesting they are associated with co-purifying RNP complexes that are not vesicle-associated. This finding is in accordance with other published data (*Shurtleff et al., 2016*; *Van Deun et al., 2014*; *Jeppesen et al., 2019*) where Ago2 is detected in the high-speed pellet but absent in the vesicle sample after more stringent purification methods are used.

To test the relevance of the three RBPs in miR-122 packaging into sEVs, we used CRISPR interference (CRISPRi) (*Gilbert et al., 2013*; *Horlbeck et al., 2016*) to systematically knock down each protein in MDA-MB-231 cells. CRISPRi promotes gene silencing by repressing transcription of the target gene. Importantly, unlike siRNA or shRNA, CRISPRi silences genes independently of the RNA-induced silencing complex (RISC). Because the RISC machinery binds to miRNAs and is responsible for miRNA-mediated gene silencing (*Bartel, 2004*), we avoided any artificial overload of the RISC machinery that might result in unpredictable effects on the miRNA sorting in our system.

Using CRISPRi, we prepared cytosols from MDA-MB-231 cells depleted of nucleophosmin and La that were then used in the in vitro packaging reactions. Both nucleophosmin and La were efficiently knocked down using this system (*Figure 4—figure supplement 2a*). However, nucleolin knock-down resulted in apparent cellular arrest with subsequent cell death. Thus, nucleolin functional analysis was not possible by knock-down. Nucleophosmin-depleted cytosol produced a mild packaging phenotype only at reduced cytosol concentrations (3 mg/ml) (*Figure 4—figure supplement 2b*). In contrast, La-depleted cytosol showed severely reduced miR-122 packaging efficiency at cytosolic concentrations of 3 and 6 mg/ml in the cell-free reactions (*Figure 4—figure supplement 2b* and *Figure 5a*). Thus, these experiments suggested a role for La in the packaging of miR-122 into vesicles formed in vitro.

## Packaging of miR-122 in vitro is La-dependent

La is an abundant RNA-binding protein that shuttles between the nucleus and cytoplasm. Within the nucleus, La helps stabilize RNA polymerase III (PolIII) transcripts by binding to polyuridine tails in their 3' termini (*Stefano, 1984*; *Rinke and Steitz, 1985*; *Wolin and Cedervall, 2002*). In the cytoplasm, La has been suggested to bind and regulate the translation of 5' terminal oligopyrimidine (TOP) mRNAs (*Intine et al., 2003*; *Cardinali et al., 2003*; *Crosio, 2000*). Moreover, in the context of cancer, La has been shown to shuttle from the nucleus to the cytoplasm at higher efficiencies in invasive cells undergoing epithelial to mesenchymal transition in order to control the translation of mRNAs bearing internal ribosome entry sites (IRES) (*Petz et al., 2012a*; *Petz et al., 2012b*). La has also been indirectly implicated in the exosomal secretion of a small Epstein-Barr viral RNA, EBV-EBER1. EBV-EBER1 interaction with La in the cytoplasm masks the viral RNA from recognition by cytoplasmic surveillance machinery and facilitates its later EV-mediated secretion (*Baglio et al., 2016*).

In order to establish whether La itself, rather than a La-associated protein, is necessary for miR-122 incorporation into EVs in vitro (*Figure 5a*), recombinant His-tagged human La was expressed and purified from insect Sf9 cells (*Figure 5—figure supplement 1*). La-depleted cytosol was reduced ~4-fold in the packaging of miR-122 but was restored to normal on addition of purified, recombinant La to the combination of membranes and La-depleted cytosol (*Figure 5b*). However, addition of La alone with or without membranes and no cytosol resulted in only background protection of miR-122 (*Figure 5b*), suggesting that additional cytosolic proteins are necessary for in vitro

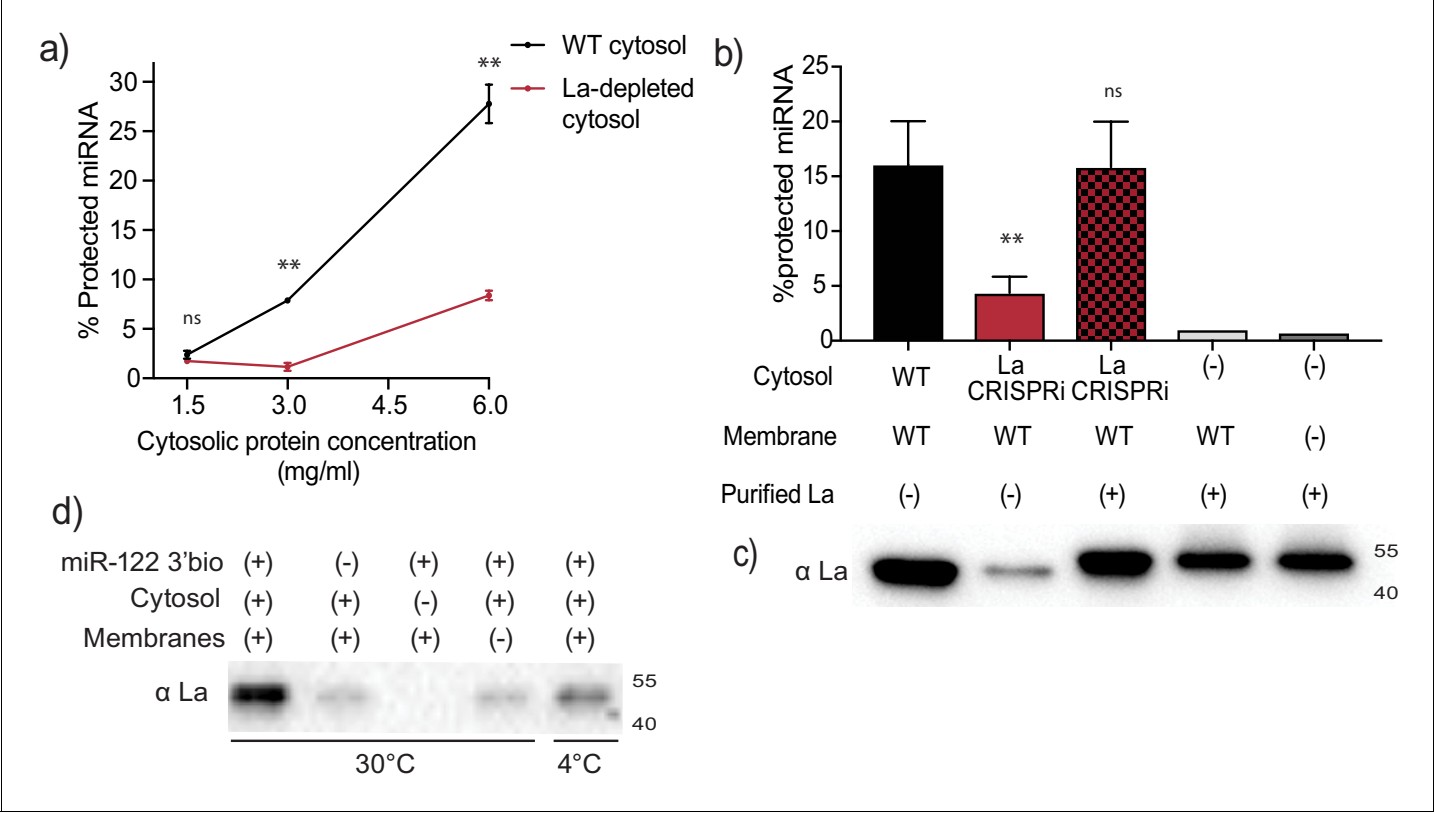

**Figure 5.** Sorting of miR-122 into EVs in vitro requires La. (**a**) Quantification of miR-122 in vitro packaging. WT and La-depleted cytosols were titrated from 1.5 to 6 mg/ml. Data plotted are from two independent experiments, each independent experiment with triplicate qPCR reactions; error bars represent standard deviation from independent samples. (**b**) Heterologously expressed La rescues miR-122 in vitro packaging. Quantification of miR-122 in vitro packaging is shown. Cytosols from WT or La-depleted cells with or without complementation using purified La were used. Reactions without cytosol, with or without membranes and with added purified La are also shown. Data plotted are from three independent experiments, each independent experiment with triplicate qPCR reactions; error bars represent standard deviation from independent samples. (**c**) Immunoblots showing the levels of endogenous or heterologously expressed added La used in the in vitro reactions as in (**b**) are shown. (**d**) Immunoblots for La following miR-122 in vitro packaging performed as for *Figure 3d* according to the conditions indicated. Statistical significance was performed using Student's t-test (**p<0.01, ns = not significant).

DOI: https://doi.org/10.7554/eLife.47544.015

The following figure supplement is available for figure 5:

**Figure supplement 1.** Purification of heterologously expressed La.

DOI: https://doi.org/10.7554/eLife.47544.016

vesicle biogenesis and/or miRNA sorting process. The level of exogenously added La protein in the in vitro reaction was adjusted to approximate the level found in an aliquot of the normal cytosol (*Figure 5c*).

We next examined the requirements for co-packaging of La and miR-122 in the cell-free reaction. An immunoblot was used to test the relative amount of La co-isolated with biotinylated miR-122 in conditions that sustain or fail to result in miR-122 packaging. We observed maximal La recovery in incubations conducted with cytosol and membranes at 30°C but much less so at 4°C (*Figure 5d*). Thus, the sorting of La bound to miR-122 requires the presence of both, membranes and cytosol and a physiological temperature of incubation.

## MiR-122 packaging into sEVs in vivo is La-dependent

Our in vitro data suggested that La was at least one of the RBPs responsible for miR-122 packaging. However, in order to confirm this data, we first examined the effects of depletion of the three RBP candidates in vivo. Previous work has demonstrated that selected miRNAs accumulate in the cytoplasm in cells deficient in RBPs required for RNA sorting into EVs (*Shurtleff et al., 2016*;

*Santangelo et al., 2016*; *Villarroya-Beltri et al., 2013*). We assessed the level of intracellular miR-122 and of another highly enriched miRNA (miR-142) upon CRISPRi depletion of each RBP candidate. Greater accumulation of miR-122 was seen upon La depletion (2.7-fold change La CRISPRi/WT) compared to nucleophosmin depletion (1.7-fold change NPM1 CRISPRi/WT) (*Figure 6a*). CRISPRi of nucleolin resulted in less cellular miR-122, which could be an indirect result of the more toxic effects of this depletion. As a control for selectivity of miRNA sorting and export, we assessed the intracellular levels of non-selectively sorted miRNAs miR-574 and miR-320a (*Figure 3d*). In contrast to the selectively sorted miRNAs, we did not observe intracellular accumulation of miR-574 and miR-320a under any of the knock-down conditions (*Figure 6a*). At least in relation to La and nucleophosmin, the depletion results in cells confirmed our previous observations on the role of these proteins in our cell-free reaction.

We next examined the relative content of miR-122 and of a non-selectively sorted miRNA, miR-320a, in sEVs in relation to their accumulation in cells depleted of La. WT and La-depleted cells had approximately the same number of purified sEVs as quantified using a Nanosight particle tracking instrument. A significant reduction in secretion of miR-122 with no change in miR-320a was accompanied by intracellular accumulation of miR-122 in the La-depleted background (*Figure 6b*). Thus, La plays a role in the selective sorting of miR-122 into sEVs, without affecting sEV biogenesis. We then tested whether La plays a broader role in the selective sorting of vHD miRNAs. We tested the secretion of validated vHD miRNAs (*Figure 2c*) that proved to reside inside vesicles (*Figure 2f*) and the secretion of non-selectively sorted miRNAs (*Figure 2c*) under La depletion vs WT conditions. All the selectively sorted miRNAs tested showed some dependence on La for their secretion, as seen by at least a 50% reduction of their secretion into sEVs in comparison to WT levels (*Figure 6c*). By contrast, all of the non-selectively sorted miRNAs tested showed little to no change under La depletion (*Figure 6c*). Thus La may play a broader role in mediating the secretion of selectively sorted miRNAs into vHD sEVs.

To test for the presence of La in sEVs isolated from cultured MDA-MB-231cells, we first purified vesicles by buoyant density flotation. The high-speed pellet was subjected to flotation in a step sucrose gradient. Buoyant membranes were permeabilized with non-ionic detergent and soluble proteins were precipitated/concentrated with acetone in order to facilitate detection by immunoblot (*Figure 6d*). We successfully detected the presence of La in the buoyant sample. Its detection was improved upon acetone precipitation/concentration (*Figure 6e*). Moreover, when we analyzed the La distribution across the iodixanol linear gradient (as in *Figure 1a*), we observed La co-fractionating with the vHD sub-population (*Figure 6—figure supplement 1*), thus confirming the observation that La is required for the selective sorting of vHD miRNAs (*Figure 6c*). Interestingly, nucleolin, one of the other RBP candidates that pulled-down with biotinylated miR-122 in the in vitro packaging reaction (*Figure 4d*) was also detected in the vHD fraction (*Figure 6—figure supplement 1*). Nucleolin may also be involved in RNA secretion of vHD transcripts.

In order to confirm that La was a soluble protein residing inside sEVs, and not simply vesicle-associated, we performed proteinase K protection assays on crude high-speed pellet material. The high-speed pellet fraction from conditioned medium was exposed to proteinase K in the presence or absence of detergent and the exposure of La was assessed by immunoblot. The La protein was resistant to proteinase K digestion and rendered sensitive upon membrane permeabilization with non-ionic detergent (*Figure 6f*). Flotillin-2, a membrane protein associated with the inner leaflet of the EV, served as a positive control and was degraded only in the presence of detergent. In contrast, Dicer, previously shown to be present in the high-speed pellet but not as an EV-encapsulated RNP (*Figure 1b*, *Figure 4—figure supplement 1*), was susceptible to degradation independent of the addition of non-ionic detergent (*Figure 6f*). We conclude that the La protein is packaged into vesicles in our cell-free reaction as well as into sEVs secreted by cells and is required for packaging of miR-122 into vesicles in vitro and in vivo.

## La directly interacts with miR-122 in vivo and in vitro

Given our results on the role of La in miR-122 sorting in cells and in the cell-free reaction, we next examined the intracellular distribution of La and the selectivity of its interaction with miR-122 versus a control miRNA that was detected, but not enriched, in vLD and vHD vesicles (miR-182; *Figure 3c*). At steady-state, La was reported to be largely confined to the nucleus (*Wolin and Cedervall, 2002*). However, other evidence supports a role for La function in the cytoplasm and its shuttling between

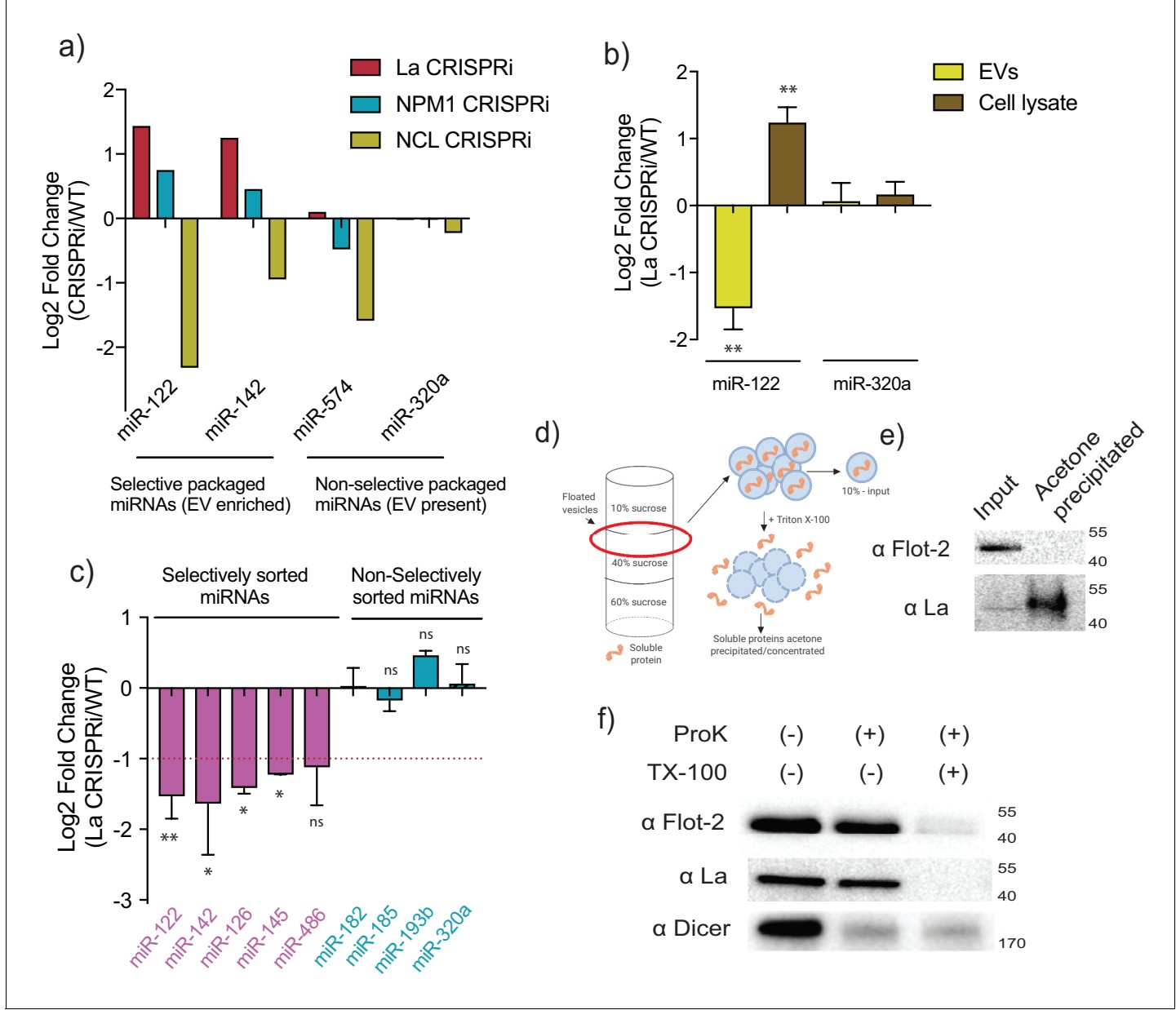

**Figure 6.** MiR-122 sorting into sEVs requires La in vivo. (**a**) Log2 fold change for intracellular levels of miRNAs of interest post-depletion of the shown RBPs by CRISPRi. MiR-122 and miR-142 were used as representatives of selectively packaged miRNAs. MiR-574 and miR-320a were used as representatives of non-selectively packaged miRNAs. Data quantified by RT-qPCR, normalized per ng of total RNA. (**b**) Log2 fold change of intracellular and secreted levels of miRNAs of interest quantified by RT-qPCR. sEVs were purified from WT and La-depleted cells. Cellular lysates were isolated at the moment of EV collection. MiR-320a was used as a control for non-selectively packaged miRNAs. Data plotted are from three independent experiments (biological replicates); each independent experiment with triplicate qPCR reactions, error bars represent standard deviation from independent samples. (**c**) Log2 fold change of secreted levels of miRNAs of interest quantified by RT-qPCR. sEVs were purified from WT and La-depleted cells. MiRNAs in pink represent miRNAs selectively sorted into vHD sEVs, miRNAs in blue represent non-selectively sorted miRNAs present in both vLD and vHD sEVs. Data plotted are from three independent experiments (biological replicates) for miR-122, miR-142 and miR-320a and from two independent experiments (biological replicates) for the rest; each independent experiment with triplicate qPCR reactions; error bars represent standard deviation from independent samples. (**d**) Schematic representation of the flotation assay and acetone precipitation as in (**e**). The high-speed pellet of conditioned medium was floated in a step sucrose gradient (input) and the soluble content was released by the addition of TX-100 and then concentrated by acetone precipitation (acetone precipitated). (**e**) sEV samples post-flotation and acetone precipitation/concentration were tested. Immunoblots for flotillin-2 and La are shown. Acetone precipitation captures mostly soluble proteins (*Feist and Hummon, 2015*), thus flotillin-2 is only detectable in the floated sample, prior to acetone precipitation, whereas La detection improves post precipitation/concentration. (**f**) Proteinase K

*Figure 6 continued on next page*

*Figure 6 continued*

protection assays in high-speed pellet fractions. Samples were treated with or without proteinase K and or Triton X-100. Immunoblots for flotillin-2, La and Dicer are shown. Statistical significance was performed using Student's t-test (*p<0.05, **p<0.01, ns = not significant).

DOI: https://doi.org/10.7554/eLife.47544.017

The following figure supplement is available for figure 6:

**Figure supplement 1.** The La protein co-fractionates with the vHD sub-population.

DOI: https://doi.org/10.7554/eLife.47544.018

the nucleus and cytoplasm (*Cardinali et al., 2003*; *Intine et al., 2003*; *Petz et al., 2012a*; *Petz et al., 2012b*). Our observation of La as an EV resident protein (*Figure 6e,f* and *Figure 6—figure supplement 1*) is consistent with at least some residence time in the cytoplasm.

Endogenous La was visualized by standard and structured illumination fluorescence microscopy (SIM) in fixed MDA-MB-231 cells. Immunofluorescence confirmed the presence of nuclear and cytoplasmic La (*Figure 7—figure supplement 1a*). The specificity of the antibody was affirmed by a significant depletion of the immunofluorescence signal in cells depleted of La by CRISPRi (*Figure 7—figure supplement 1a,b*). We observed that most of the La puncta were smaller than ~100 nm, but when bigger La structures were present (>500 nm), they could often be observed co-localizing with late endosomal marker Rab7 (*Figure 7—figure supplement 2a*). This co-localization was further explored at a higher level of resolution, SIM (*Figure 7—figure supplement 2b*). Standard fluorescence microscopy and SIM revealed cytoplasmic puncta of La, some of which co-localized with the late endosomal marker, Rab7 (*Figure 7—figure supplement 2a,b*).

We evaluated the interaction of La with two miRNAs by co-immunoprecipitation from mechanically ruptured MDA-MB-231 cells. A post-nuclear supernatant fraction of cells was mixed with La antiserum and the IP sample was split for protein and RNA analysis by immunoblot and RT-qPCR, respectively (*Figure 7a*). Approximately 15% of the La protein and miR-122 was precipitated (*Figure 7b and c*). MiR-182, which is present but not enriched in vLD and vHD sEVs, was used as a negative control and did not co-immunoprecipitate with La (*Figure 7c*).

In order to explore the interaction of La and miR-122 directly, we used purified, recombinant La incubated with miR-122 and evaluated the binding affinity by electrophoretic mobility shift assays (EMSAs). Purified La was titrated, mixed with 5' fluorescently-labeled miR-122 and incubated at 30° C. After incubation, complexes were separated by electrophoresis and detected by in-gel fluorescence (*Figure 7d*). As negative and positive controls for La interaction we used 22 nt RNAs comprised of alternating purines or polyuridine, respectively (*Figure 7—figure supplement 3a and b*). The measured $K_d$ for La:miR-122 was 4.8 nM (*Figure 7e*). Notably, this measurement indicates that the affinity of miR-122 for La is greater than the affinity previously reported for Ago2 and miRNAs, with $K_d$s from 10 to 80 nM (*Tan et al., 2009*). We therefore conclude that the La:miR-122 complex displays high affinity and specific interaction in vitro.

## Finding the miR-122 motif responsible for its packaging

Lastly, we explored the miR-122 motif responsible for its association with La and packaging into sEVs in the cell-free reaction. La has been shown to bind to UUU sequences at the 3' end of PolIII transcripts (*Stefano, 1984*) and miR-122 contains a UUUG at that terminus (*Figure 8a*). We designed a variant of miR-122 where A residues replaced the 3'U bases and found that this species is poorly packaged into vesicles in the cell-free reaction (3% efficient incorporation vs. 15% for WT) and displays a greatly reduced (~100-fold) affinity for La in the EMSA analysis (WT $K_d$ 4.8 nM vs 3'mut $K_d$ 538 nM) (*Figure 8a,b and c* and *Figure 8—figure supplement 1a*). We conclude that the UUU sequence at the 3' end of miR-122 is necessary for a high affinity interaction with La, which may correlate to the requirement for these residues in the packaging of miR-122 into vesicles in the cell-free reaction.

To identify possible motifs beyond the literature-based UUU candidate, we also performed a multiple expectation maximization (EM) for motif elicitation (MEME) analysis for 49 vHD EV miRNAs. We considered miRNAs that were at least 10-fold enriched, as detected by profiling and sequencing, as well as those unique miRNAs in the vHD sub-population that were detected by sequencing. This approach identified 'UGGA' as an EV-enriched miRNA motif. This motif was present in 13 (26%) of

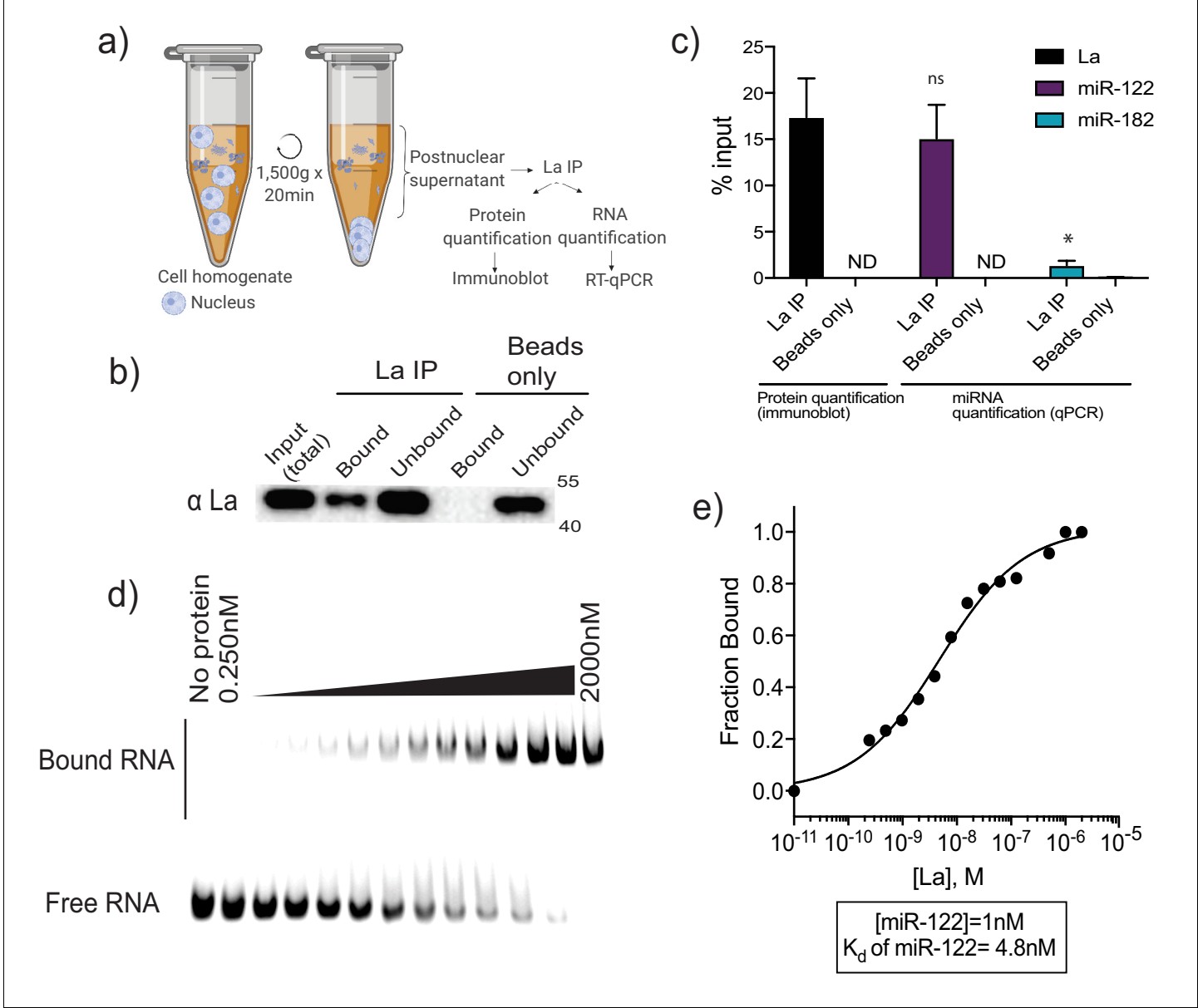

**Figure 7.** La interacts with miR-122 in vitro and in vivo. (**a**) Schematic representing immunoprecipitation of endogenous La shown in (**b**) and (**c**). (**b**) Immunoblot for La post-La IP. (**c**) Quantification of the La immunoblot post La IP. The relative levels, as percentage of input, of miR-122 and miR-182 co-IP with La are also shown. MiR-182, a non-selectively sorted miRNA, served as a negative control for La binding. Data plotted are from two independent experiments (biological replicates), for qPCR data each independent experiment with triplicate qPCR reactions; error bars represent standard deviation from independent samples. (**d**) EMSA assays using 5′ fluorescently labeled miR-122. Purified La was titrated from 250 pM to 2 μM. In gel fluorescence was detected. (**e**) Quantification of (**d**) showing the calculated $K_d$. Fraction bound was quantified as a function of exhaustion of free miRNA. Statistical significance was performed using Student's t-test (*$p < 0.05$, ns = not significant).

DOI: https://doi.org/10.7554/eLife.47544.019

The following figure supplements are available for figure 7:

**Figure supplement 1.** Detection of nuclear and cytoplasmic La by immunofluorescence.

DOI: https://doi.org/10.7554/eLife.47544.020

**Figure supplement 2.** La associates with Rab7-positive vesicles.

DOI: https://doi.org/10.7554/eLife.47544.021

**Figure supplement 3.** Controls for La specificity during EMSA.

DOI: https://doi.org/10.7554/eLife.47544.022

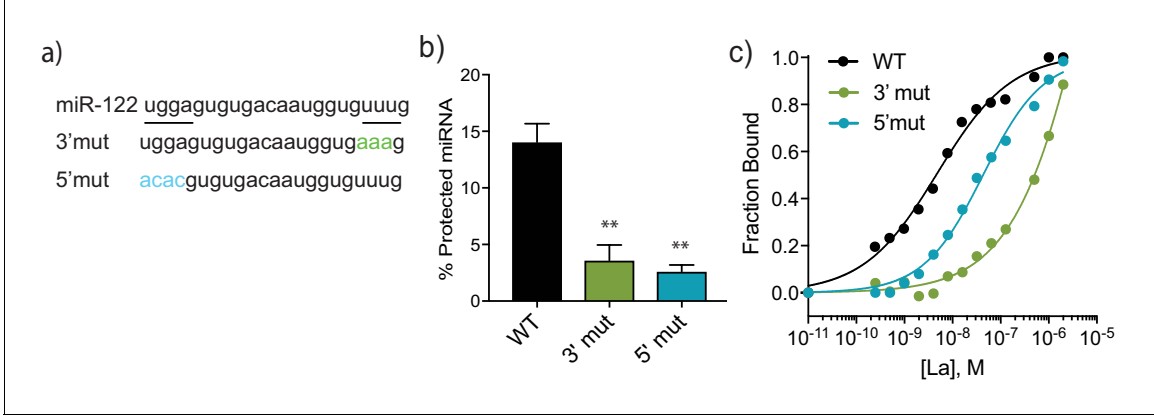

**Figure 8.** A bipartite motif in miR-122 is required for its packaging and interaction with La in vitro. (a) The sequences of miR-122 WT and mutated versions are shown. (b) Quantification of in vitro packaging of miR-122 WT and mutated versions. Data plotted are from three independent experiments, each with triplicate qPCR reactions; error bars represent standard deviation from independent samples. (c) Binding affinity curves (as calculated from EMSAs) for miR-122 and the mutated versions are shown. Fraction bound was quantified as a function of exhaustion of free miRNA. Statistical significance was performed using Student's t-test (**p<0.01).

DOI: https://doi.org/10.7554/eLife.47544.023

The following figure supplement is available for figure 8:

**Figure supplement 1.** EMSA for miR-122 mutated versions.

DOI: https://doi.org/10.7554/eLife.47544.024

the 49 analyzed miRNAs, including the 5' sequence of miR-122 (*Figure 8a*), and it was by far the most overrepresented motif detected by this analysis. The UGGA sequence appeared to be sEV specific, as it did not emerge when similar analyses were performed with exclusively cellular miRNAs (miRNAs detected in cellular lysates but absent in the sEV samples). We then manually assessed the pool of exclusively cellular miRNAs for this motif where it appeared in 5 out of 97 analyzed miRNAs (5%). A variant of this 5' sequence motif impaired miR-122 incorporation in the in vitro packaging reaction (*Figure 8b*) and decreased the affinity ~10-fold for La compared to the WT sequence (*Figure 8c* and *Figure 8—figure supplement 1b*). Thus, we conclude that a bipartite, or possibly even more complex motif directs recognition of miR-122 by La and consequently results in its selective sorting into EVs and secretion from MDA-MB-231 cells.

## Discussion

### Different molecular mechanisms of miRNA sorting

Here we document the separation of two sEV populations, likely of different sub-cellular origin, which appear to employ different molecular mechanisms of miRNA sorting (*Figure 9*). We found that sEVs with a density ranging from 1.09 to 1.11 g/ml (vLD) may originate in the plasma membrane, whereas the sEVs with a density ranging from 1.14 to 1.16 g/ml (vHD) are more characteristic of exosomes, originating in the endocytic pathway. Some miRNAs are non-selectively sorted into both sEV sub-populations, as shown by their similar relative abundances in the cellular lysate and both sEV sub-populations (*Figure 3d*). We observed a slight enrichment for non-selectively sorted miRNAs in the vHD sub-population (*Figure 3a*) and a minor depletion of non-selectively sorted miRNAs in the vLD sub-population (*Figure 3b*).

Non-selective sorting may correlate to the fraction of miRNAs unbound to RISC. Association of miRNAs with RISC inside cells varies greatly: >1,800-fold for different miRNAs present in the same cell line (*Flores et al., 2014*). As a result, some miRNAs may exist in the cytoplasm as mostly RISC-free species. Our data show that Ago2 and Dicer are not present in sEVs derived from MDA-MB-231 cells (*Figure 1b* and *Figure 4—figure supplement 1*). Thus, we suggest that the non-selectively

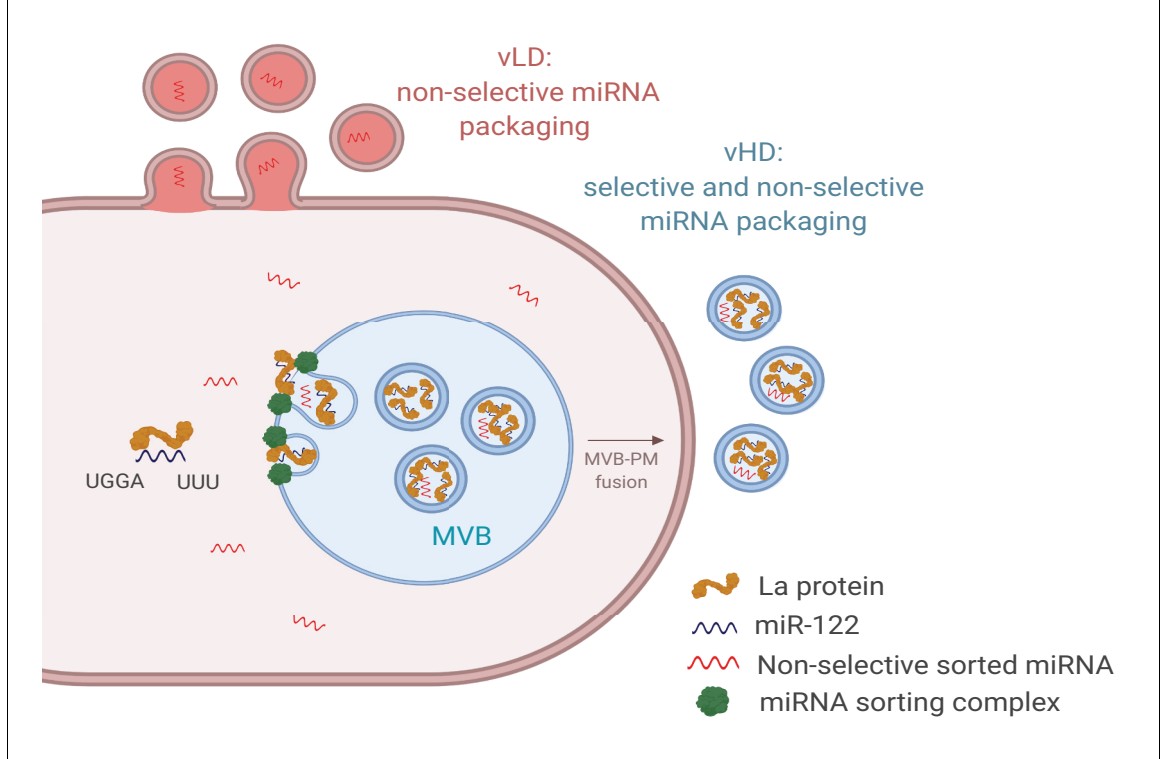

**Figure 9.** Diagram representing the current model of miRNA sorting into extracellular vesicles derived from MDA-MB-231 cells. At least one vesicular species within the vHD sub-population has its origin in the endocytic pathway, representing the classical exosomes. We propose a selective mechanism of miRNA sorting occurring at the site of vHD vesicle biogenesis where La, along with bound miR-122 (which interaction is mediated by at least two RNA motifs), is targeted for capture into a bud invaginating into the interior of an endosome. vLD vesicles may originate at the plasma membrane, representing shedding vesicles. There may be no selective miRNA sorting occurring at the site of vLD vesicle biogenesis. Non-selective miRNA sorting occurs in both the vHD and vLD sub-populations. MVB: multivesicular body, PM: plasma membrane.

DOI: https://doi.org/10.7554/eLife.47544.025

sorted miRNAs may represent the cytosolic pool of RISC-free miRNAs, not actively engaged in silencing, and their presence in sEVs is the result of a passive mechanism of sorting.

P-bodies have been shown to be closely associated with MVBs (*Gibbings et al., 2009*) and the levels of target mRNAs can influence the levels of miRNA association with RISC (*Flores et al., 2014*), as well as miRNA secretion as detected in the high-speed pellet fraction sedimented from conditioned media (*Squadrito et al., 2014*). Thus, miRNAs released from P bodies and freed from RISC may be proximal to endosomes engaged in the production of vHD vesicles and distant from the site of vLD vesicle biogenesis, thus resulting in a slight enrichment and slight depletion of these miRNAs in the vHD and vLD vesicles, respectively.

We focused the bulk of our investigation on a set of miRNAs selectively packaged in the vHD sub-population. These selectively sorted miRNAs may belong to the pool of pre-existing RISC-free miRNAs, or may be displaced from RISC prior to or concomitant with sorting into vHD vesicles. Importantly, we found that this mechanism of selective miRNA sorting is specific for the vHD sub-population. We speculate that the sorting machinery required to enrich certain miRNAs in exosomes is specific for MVBs, and it is absent at the site of origin of the vLD vesicles. However, enveloped viruses that bud from the cell surface clearly do sort their RNA genomes in a selective manner (*D'Souza and Summers, 2005*; *Swanson and Malim, 2006*) and the same may be true of other RNA biotypes in the vLD sub-population, especially in other cell lines. We previously showed that tRNAs with an unknown post-transcriptional modification in the D-loop are highly enriched in sEVs in comparison to the cells from the same culture (*Shurtleff et al., 2017*). Similarly, we detected apparently similarly modified tRNA species in both sEV sub-populations secreted from MDA-MB-231 cells

(*Figure 3—figure supplement 2e*). The machinery for the active sorting of the modified tRNA may be shared for both sEV sub-populations.

It is widely suggested that sEV miRNAs play a role in intercellular communication (*Fong et al., 2015*; *Zhou et al., 2014*; *Tominaga et al., 2015*; *Rana et al., 2013*; *Hsu et al., 2017*). Thus, purifying sEV sub-populations, with differential miRNA content, is critical in order to accurately assess their role in intercellular communication. One study purified different sEVs based on their buoyant densities and showed that the vLD sub-population triggered the differential expression of 257 genes in recipient cells, whereas the vHD sub-population caused the differential expression of 1116 genes (*Willms et al., 2016*). Although this study did not determine if any gene expression differences were the result of miRNA transfer, the effect could be partially explained by the distinct miRNA composition of the two sEV sub-populations. Further studies using purified sEV sub-populations are needed to understand the physiological consequence of selective vs. non-selective sorting mechanisms. Additionally, it needs to be determined if the effects are directly attributable to miRNAs or some other constituents of the two sEV populations.

## The role of La in selective sorting of miRNAs

Here we show in vitro and in vivo evidence for the requirement of La for packaging of miR-122 and other selectively sorted miRNAs into vHD sEVs derived from MDA-MB-231 cells. The La protein is an abundant RBP that is upregulated in some cancers (*Sommer et al., 2011b*; *Trotta et al., 2003*; *Al-Ejeh et al., 2007*; *Sommer et al., 2011a*). Its presence in the cytoplasm has been linked to the translational regulation of mRNAs bearing IRES (*Petz et al., 2012a*; *Petz et al., 2012b*), as well as 5'TOP mRNAs (*Intine et al., 2003*; *Cardinali et al., 2003*; *Crosio, 2000*). Interestingly, it was previously reported that 5'TOP mRNAs are enriched in sEVs derived from HEK293T cells in comparison to their relative cellular levels (*Shurtleff et al., 2017*), although we have not yet evaluated the abundance of 5'TOP mRNAs in sEVs isolated from MDA-MB-231 cells.

Little is known about how cytoplasmic RNA binding proteins, such as La, along with bound RNA molecules are targeted and recognized for secretion. The phosphorylation status of La has been suggested to be important for its intracellular localization (*Intine et al., 2003*). In addition, sumoylation of La has been shown to be important for La-dependent mRNA transport in neurons (*van Niekerk et al., 2007*) and sumoylation of another RNA-binding protein in sEVs (hnRNPA2B1) has been shown to be important for export (*Villarroya-Beltri et al., 2013*). Post-translational modifications such as phosphorylation, ubiquitylation and sumoylation may serve to direct RNPs to the ESCRT machinery and thus into membrane buds that are internalized into MVBs.

The La protein binds to and stabilizes poly-uridine stretches of PolIII transcripts that mature in the nucleus (*Wolin and Cedervall, 2002*). Likewise in the cytoplasm, La may serve a similar function in the stabilization of polyuridine tracks in 5'TOP mRNAs (*Intine et al., 2003*). Here we show that La directly binds to, and promotes the membrane enclosure of miR-122 in a cell-free reaction. We obtained evidence that a 3' polyuridine sequence of miR-122 is required for this interaction. This role of La may extend to miRNAs secreted by other cell lines. B cell-derived sEVs are enriched for 3' uridylylated miRNA isoforms in comparison to their intracellular counterparts (*Koppers-Lalic et al., 2014*), and La has been detected by mass spectrometry in the high-speed pellet fraction from B cell conditioned medium (*Meckes et al., 2013*).

In addition to the known poly (U) binding site, sequence motif analysis of the most highly enriched miRNAs in the vHD vesicles isolated from MDA-MB-231 conditioned medium identified a second, previously unreported motif required for La:miR-122 high affinity interaction and sorting into sEVs in our cell-free reaction. A 5' UGGA sequence resembles the motif shown to be required for miRNA sorting into hepatocyte sEVs, dependent on the SYNCRIP protein (*Santangelo et al., 2016*). However, we did not detect SYNCRIP in our mass spectrometry analysis for RBP partners responsible for miR-122 sorting in vitro (*Figure 4d*). Thus, it is possible hepatocytes and breast cancer epithelial cells differ in their mechanisms of miRNA sorting. The presence of both motifs, UUU and UGGA, in miR-122 may allow for an exceptionally high affinity interaction with La. The measured $K_d$ for the La:miR-122 complex is 4.8 nM, indicating tighter binding than that reported for Ago2 to miRNAs using a similar assay (*Tan et al., 2009*). This high affinity interaction may explain how selected miRNAs are sequestered into exosomes away from RISC.

As we have shown with La, other RBPs (YBX1 [*Shurtleff et al., 2016*], hnRNPa2b1 [*Villarroya-Beltri et al., 2013*] and SYNCRIP [*Santangelo et al., 2016*]) are involved in the selective sorting and

secretion of miRNAs in different cell lines. Conceivably, this process could be organized and controlled in a cell-specific manner by the assembly of RNA granules including other more abundant RNAs such as tRNA, Y-RNA and vault RNA (*Shurtleff et al., 2017*). In this respect, we find that most of the La protein in the cytoplasm is localized in punctate structures, some of which closely associate with the late endosomal marker Rab7 (*Figure 7—figure supplement 2a,b*). It will be instructive to examine the cytoplasmic co-localization of La with the other RBPs implicated in miRNA secretion in EVs.

## Significance of selective miRNA sorting into EVs in cancer

Intercellular communication mediated by miRNAs is a subject of considerable interest, but in most cases, it remains unproven. Many studies using cancer cells suggest EV-mediated miRNA transfer to recipient cells (*Fong et al., 2015*; *Zhou et al., 2014*; *Tominaga et al., 2015*; *Rana et al., 2013*; *Hsu et al., 2017*), possibly to promote metastasis. In the context of breast cancer for example, sEV resident miR-122 has been suggested to promote metastasis through reprogramming glucose metabolism in the resident cells of the premetastatic niche (*Fong et al., 2015*). In another example, sEV-associated miR-105 may promote invasiveness in metastatic breast cancer by disruption of tight junctions, thus promoting vascular permeability and tumor cell invasion (*Zhou et al., 2014*). For sEV miRNAs to be functionally transferred to recipient cells, fusion of the EV membrane with a recipient membrane (either plasma membrane or endosomal membrane) must precede the presentation of an exogenous miRNA to the target cell cytoplasm. We find that the highly-selected miRNAs in exosomes are not bound to Ago2. Thus, the usual function of a miRNA would require loading from the EV interior to a cytoplasmic RISC. Unfortunately, nothing is known about the cytoplasmic presentation of miRNAs derived from EVs.

Alternatively, tumor cell invasiveness may depend on the selective secretion and disposal of tumor suppressor miRNAs. Such a suggestion has been made for miR-23b (*Ostenfeld et al., 2014*) and miR-100 (*Cha et al., 2015*), which are secreted at high efficiency by cancer cells. These two mechanisms, intercellular communication and disposal, are not mutually exclusive. It is possible that sEVs containing detrimental miRNAs have been repurposed to serve as shuttles for intercellular communication, as has been suggested for the secretion of miR-142 (*Dickman et al., 2017*). Of relevance to the work we report here, miR-122 has been suggested as a tumor suppressor with cells developing greater invasiveness on loss of this miRNA (*Coulouarn et al., 2009*; *Wang et al., 2012*). Nonetheless, the importance of sEV resident miR-122 as a molecule to convey a message to promote metastasis has also been suggested (*Fong et al., 2015*).

A bigger challenge in the field of cancer and EV biology is that claims for a role of EV-associated miRNAs are based on crude preparations of sedimentable particles or co-culture experiments. Here we show that Ago2 is secreted to the extracellular space as a RNP, not vesicle-associated, that co-isolates with sEVs in the high-speed pellet. Thus, clear demonstration that the implicated miRNAs in cancer development are enclosed within a membrane vesicle is still missing. Studies purifying, characterizing, understanding the cellular biology behind different sEV sub-populations and their functional effects in the context of cancer will prove important in the future.

# Materials and methods

**Key resources table**

| Reagent type (species) or resource | Designation | Source or reference | Identifiers | Additional information |
|---|---|---|---|---|
| Gene (*Homo sapiens* - Ssb) | La | Addgene | RRID:Addgene_38241 | N/A |
| Antibody | anti-La (mouse monoclonal) | Origene Technologies | RRID: AB_2239866 | IF 1:100. WB: 1:1000 |
| Antibody | anti-Rab7 (goat polyclonal) | Santa Cruz | RRID: AB_2175486 | IF 1:100 |

*Continued on next page*

Continued

| Reagent type (species) or resource | Designation | Source or reference | Identifiers | Additional information |
|---|---|---|---|---|
| Antibody | anti-CD63 (mouse monoclonal) | Developmental Studies Hybridoma Bank | RRID: AB_528158 | IF 1:100 |
| Antibody | anti-CD9 (mouse monoclonal) | Biolegend | RRID: AB_314907 | IF 1:100 |
| Antibody | Anti-ITGA3 (rabbit polyclonal) | Abcam | RRID: AB_2810979 | WB 1:1000 |
| Antibody | anti-MFGE-8 (mouse monoclonal) | R and D systems | RRID: AB_2142466 | WB 1:1000 |
| Antibody | anti-Flot-2 (mouse monoclonal) | BD biosciences | RRID: AB_397766 | WB 1:1000 |
| Antibody | anti-Tsg101 (mouse monoclonal) | GeneTex | RRID: AB_373239 | WB 1:1000 |
| Antibody | anti-CD63 (mouse monoclonal) | BD biosciences | RRID: AB_396297 | WB 1:1000 |
| Antibody | anti-Ago2 (rabbit monoclonal) | Cell signaling | RRID: AB_2096291 | WB 1:1000 |
| Antibody | anti-Ago2 (rabbit monoclonal) | Abcam | RRID: AB_2713978 | WB 1:1000 |
| Antibody | anti-dicer (rabbit polyclonal) | Santa cruz | RRID: AB_639122 | WB 1:1000 |
| Antibody | anti-vinculin (rabbit monoclonal) | Abcam | RRID: AB_11144129 | WB 1:1000 |
| Antibody | anti-Rab27a (mouse monoclonal) | Abcam | RRID: AB_945112 | WB 1:1000 |
| Antibody | anti-nucleophosmin (mouse monoclonal) | Abcam | RRID: AB_297271 | WB 1:1000 |
| Antibody | anti-nucleolin (rabbit polyclonal) | Abcam | RRID: AB_776878 | WB 1:1000 |
| Antibody | anti-Hsc70 (rabbit monoclonal) | Abcam | RRID: AB_444764 | WB 1:1000 |
| Antibody | anti-tubulin (mouse monoclonal) | Abcam | RRID: AB_2241126 | WB 1:1000 |
| Cell line (*Spodoptera frugiperda*) | Sf9 | Other | N/A | Cell culture facility at UC Berkeley |
| Cell line (*Homo sapiens*) | MDA-MB-231 | Other | N/A | Cell culture facility at UC Berkeley |
| Cell line (*Homo sapiens*) | MDA-MB-231 Rab27a KO | This study | N/A | Obtained by CRISPR-Cas9 |
| Cell line (*Homo sapiens*) | MDA-MB-231 Rab35 KO | This study | N/A | Obtained by CRISPR-Cas9 |
| Recombinant DNA reagent | UCOE- EF1a-dCas9-BFP-KRAB (plasmid) | N/A | N/A | Dr. Jonathan Weissman (University of California, San Francisco) |
| Recombinant DNA reagent | EF1Alpha-puro-T2A-BFP (plasmid) | Addgene | RRID:Addgene_60955 | N/A |
| Software, algorithm | GO term finder | N/A | RRID:SCR_008870 | N/A |
| Software, algorithm | REVIGO | N/A | RRID:SCR_005825 | N/A |

*Continued on next page*

*Continued*

| Reagent type (species) or resource | Designation | Source or reference | Identifiers | Additional information |
|---|---|---|---|---|
| Software, algorithm | mirDeep | N/A | RRID:SCR_010829 | N/A |
| Software, algorithm | HISAT2 | N/A | RRID:SCR_015530 | N/A |
| Software, algorithm | Bowtie | N/A | RRID:SCR_005476 | N/A |
| Software, algorithm | Samtools | N/A | RRID:SCR_002105 | N/A |
| Software, algorithm | Bedtools | N/A | RRID:SCR_006646 | N/A |
| Peptide, recombinant protein (enzyme) | TGIRT-III Enzyme | InGex | N/A | See Materials and methods for further details. |

## Cell lines, media and general chemicals

MDA-MB-231 cells were cultured in DMEM with 10% FBS (Thermo Fisher Scientific, Waltham, MA). MDA-MB-231 cells were confirmed by short tandem repeat profiling (STR) and tested negative for mycoplasma contamination. For EV production, cells were seeded at ~10% confluency in 10% exosome-depleted FBS (System Biosciences, Palo Alto, CA) in 150 mm CellBIND tissue culture dishes (Corning, Corning, NY) containing 30 ml of growth medium. EVs were collected when cells reached ~80% confluency (~72 hr). Unless otherwise noted, all chemicals were purchased from Sigma Aldrich (St. Louis, MO).

## Extracellular vesicle purification

Conditioned medium (720 ml for TGIRT-seq and miRNA profiling and 420 ml for all other experiments) was harvested from 80% confluent MDA-MB-231 cultured cells. All subsequent manipulations were performed at 4°C. Cells and large debris were removed by centrifugation in a Sorvall R6+ centrifuge (Thermo Fisher Scientific) at 1000xg for 15 min followed by 10,000xg for 15 min in 500 ml vessels using a fixed angle FIBERlite F14−6 × 500 y rotor (Thermo Fisher Scientific). The supernatant fraction was then centrifuged at ~100,000 xg (28,000 RPM) onto a 60% sucrose cushion in buffer C (10 mM HEPES pH 7,4, 0.85% w/v NaCl) for 1.5 hr using two SW-28 rotors. The interface over the sucrose cushion was collected and pooled for an additional ~130,000 xg (32,500 RPM) centrifugation over a 60% sucrose cushion in a SW41 rotor for 15 hr. The collected interface from the first sucrose cushion should not exceed a sucrose concentration of 21%, as measured by refractometry, for the second centrifugation in the SW41 to be successful. Higher concentrations of sucrose cause the EVs to equilibrate at the ambient buoyant density, impeding sedimentation. For purification of EV sub-populations based on their distinct buoyant density, the cushion-sedimented vesicles were collected and mixed with 60% sucrose to a final volume of 4 ml (sucrose final concentration ~40%). Layers of 1.5 ml of 25%, 20%, 15%, 10% and 5% iodixanol (Optiprep) solutions in buffer C were sequentially overlaid and samples were centrifuged at ~160,000 xg (36,000 RPM) for 15 hr in a SW41 rotor. Fractions (400 ul) from top to bottom were then collected and mixed with Laemmli buffer for immunoblot analysis or RNA was extracted using a mirVana miRNA isolation kit (Thermo Fischer Scientific). In some cases, such as the detection of CD63, La or nucleolin, the floated fractions were centrifuged in a SW41 rotor at ~130,000 g in order to concentrate them and improve their detection. This method produced a linear density gradient from 1.036 to 1.24 g/ml (*Figure 1—figure supplement 1—source data 1*).

For the initial detection of Ago2 and Dicer (as in *Figure 4—figure supplement 1*), vesicles were not sedimented over a 60% sucrose cushion. Sedimented EVs from the SW28 centrifugation were resuspended in 800 ul of 30% iodixanol and layers of 800 ul of 25%, 20%, 15%, 10% and 5% iodixanol were layered on top and centrifuged in a SW55 rotor at ~140,000 xg (38,000 RPM) for 15 hr. Fractions (200 ul) were collected from top to bottom and mixed with Laemmli buffer for immunoblots analysis.

For EV purification in bulk (without discriminating among EV sub-populations), the EVs collected after the first SW41 centrifugation were mixed with 60% sucrose to a final volume of 9 ml (sucrose concentration ~50%). Layers of 1.5 ml of 40% and 10% sucrose were overlaid and the sample was centrifuged at ~160,000 xg (36,000 RPM) for 15 hr in a SW41 rotor. Buoyant vesicles spanning from the 40% sucrose region and the 10/40% interface were collected and mixed with Laemmli buffer or RNA was extracted with a mirVana miRNA isolation kit. For further processing to improve protein detection, buoyant vesicles were first diluted in buffer C (10 mM HEPES pH 7,4, 0.85% w/v NaCl) to a final volume of 5 ml and soluble content was released by adding 10% Triton X-100 (TX-100) to a final concentration of 1%. Samples were then homogenized by the use of a vortex mixer for 1 min, cooled on ice for 10 min and homogenized again for an extra min. To precipitate proteins, we added 4 volumes of cold acetone (previously stored at −20°C) and the mixture was incubated at −20°C overnight. Precipitated proteins were collected using a Sorvall RC 6+ centrifuge by centrifugation at 16,000xg (11,600 RPM) for 30 min at 4°C in a FIBERlite F211−8 × 50Y rotor (Thermo Fischer Scientific). The pellet fraction was then resuspended in Laemmli buffer for analysis by immunoblotting.

For proteinase K protection assays, the EVs were collected by centrifugation at ~100,000 xg (28,000 RPM) for 1.5 hr using two SW-28 rotors. Pellet fractions resuspended in buffer C were pooled and centrifuged at ~160,000 xg (36,000 RPM) in a SW55 rotor for 1 hr. The pellet was resuspended in buffer C and split into three equal aliquots. One sample was left untreated, another sample treated with 5 ug/ml proteinase K on ice for 20 min, and the last was mixed with TX-100 to a final concentration of 1% prior to proteinase K treatment. Proteinase K was inactivated with 5 mM phenylmethane sulfonyl fluoride (PMSF) (5 min on ice) and samples were then mixed with Laemmli buffer for immunoblot analysis.

## Mass spectrometry analysis of EV sub-populations

EV sub-populations purified from the iodixanol gradient (as described above) were diluted with buffer C to a final volume of 30 ml. The diluted samples were then centrifuged at ~100,000 xg (28,000 RPM) in a SW28 rotor for 1.5 hr. Pellet samples were resuspended in Laemmli buffer without bromophenol blue and electrophoresed in a 4–20% acrylamide Tris-Glycine gradient gel (Life Technologies) for ~3 min. The bulk of proteins were stained with Coomassie and the stained band was excised from the gel with a new razor blade. Samples were submitted to the Vincent J. Coates Proteomics/Mass Spectroscopy laboratory at UC Berkeley for in-gel tryptic digestion of proteins followed by liquid chromatography and mass spectrometry analysis according to their standards.

For sub-cellular localization analysis of the detected proteins, the list of proteins detected by mass spectrometry was analyzed first using the GoTermFinder developed at the Lewis-Sigler Institute at Princeton (*Boyle et al., 2004*) followed by REVIGO analysis (*Supek et al., 2011*) to obtain the scatterplots of sub-cellular localization along with their organelle association.

## Nanoparticle tracking analysis

Extracellular vesicles purified on linear iodixanol gradients were diluted 1:100 with PBS filtered with a 0.02 um filter (Whatman GmbH, Dassel, Germany). The mixture was drawn into a 1 ml syringe and inserted into a Nanosight LM10 instrument (Malvern, UK). Particles were tracked for 60 s using Nanosight nanoparticle tracking analysis software. Each sample was analyzed five times and the counts were averaged.

## Immunoblots

Exosomes were prepared by mixing sedimented vesicles with 1X Laemmli buffer, or 1X Laemmli buffer without DTT when detection of CD63 was performed. Cell lysates were prepared by adding lysis buffer (10 mM Tris-HCl, pH 7.4, 100 mM NaCl, 0.1% sodium dodecyl sulfate, 0.5% sodium deoxycholate, 1% Triton X-100, 10% glycerol) to cell pellets. Protein was quantified using a BCA Protein Assay Kit (Thermo Fischer Scientific), and the selected amount was mixed with Laemmli buffer. Samples were heated at 95°C for 5 min and separated on 4–20% acrylamide Tris-Glycine gradient gels (Life Technologies). Proteins were transferred to polyvinylidene difluoride membranes (EMD Millipore, Darmstadt, Germany), blocked with 5% bovine serum albumin in TBST and incubated overnight with primary antibodies. Blots were then washed with TBST, incubated with anti-rabbit or anti-mouse secondary antibodies (GE Healthcare Life Sciences, Pittsbugh, PA) and detected with

ECL-2 reagent (Thermo Fisher Scientific). Primary antibodies used in this study were anti-CD9 #13174S (Cell Signaling Technology, Danvers, MA), anti-alpha integrin alpha 3 #ab190731 (Abcam, Cambridge, MA), anti-Mfg-e8 #MAB2767 (R and D systems, Minneapolis, MN), anti-flotillin-2 #610383 (BD Biosciences, San Jose, CA), anti-TSG101 #GTX70255 (Genetex, Irvine, CA), anti-CD63 #BDB556019 (BD Biosciences), anti-Ago2 #2897 (Cell Signaling Technology), anti Ago2 ab#186733 (abcam), anti-Dicer #sc-30226 (Santa Cruz Biotechnology), anti-Lupus La #TA500406 (Origene Technologies, Rockville, MD), anti-vinculin #ab129002 (Abcam), anti-Rab27a ab55667 (Abcam), anti-nucleophosmin1 #ab10530 (Abcam), anti-nucleolin #ab22758 (Abcam), anti-tubulin #ab7291 (Abcam), and anti-Hsc-70 #ab19136 (Abcam).

## Quantitative real-time PCR

Cellular and EV RNAs were extracted with a mirVana miRNA isolation kit (Thermo Fischer Scientific), unless otherwise specified. Taqman miRNA assays for miRNA detection were purchased from Life Technologies. Assay numbers were: hsa-miR-193b, #002366; hsa-miR-29b-1, #002165; hsa-miR-486, #001278; hsa-miR-574–3 p, #002349; hsa-miR-320a, #002277; hsa-miR-142–3 p, #000464; hsa-miR-126, #000451; hsa-mir-145–5 p, #002278; hsa-mir-122–5 p, #002245; hsa-mir-451a, #001141; hsa-mir-182–5, 002334; hsa-miR-429, #001024; hsa-miR-675, #121124_mat; hsa-miR-193a-5p, #002281, hsa-miR-185 #002271. As there are no well-accepted endogenous control transcripts for EVs, the relative quantification was normalized to equal amounts of starting RNA material. RNA was quantified by an Agilent 2100 Bioanalyzer (Agilent Technologies, Santa Clara, CA) according to the manufacturer's instructions. RNA (1 ng) was used for reverse transcription according to the manufacturer's instructions. Relative quantification was calculated from the expression $2^{-(Ct_{(control)}-Ct_{(experimental)})}$. Taqman qPCR master mix with no amperase UNG was obtained from Life Technologies and quantitative real-time PCR was performed using an ABI-7900 real-time PCR system (Life Technologies).

## Immunofluorescence

MDA-MB-231 cells on 12 mm round coverslips (Corning) were fixed by adding 4% EM-grade formaldehyde (Electron Microscopy Sciences, Hatfield, PA) for 20 min at room temperature. Subsequently, cells were washed three times with PBS and permeabilized/blocked by adding blocking buffer (0.1% TX-100 in 2 FBS% for La validation experiments as well as for CD63 and CD9 immunofluorescence, or 0.02% saponin in 2% FBS for superresolution structured illumination microscopy for La and Rab7 analysis (SIM)) for 20 min. Cells treated with 0.02% saponin retain intact nuclear envelopes and endoplasmic reticulum membranes (*Sirkis et al., 2017*; *Gorur et al., 2017*), which allowed us to visualize cytoplasmic La. Cells were then incubated with 1:100 dilution of anti-La (Origene Technologies, #TA500406), 1:100 dilution of anti-Rab7 (Santa Cruz, sc-6563), 1:100 dilution of CD9 (Biolegend, #312102) and 1:100 dilution of CD63 (DSHB, #H5C6) in blocking buffer for 1.5 hr at room temperature, extensively washed and incubated in secondary antibodies diluted 1:1000 in blocking buffer and Alexa Fluor 488 and 647 (Thermo Fischer Scientific), for 1.5 hr. Cells were extensively washed, rinsed briefly in $dH_2O$ and mounted on slides with ProLong Gold with DAPI (Thermo Fischer). For validation experiments, WT and La-depleted cells were imaged keeping all the settings constant in an Axio Observer Z1 (Zeiss, Oberkochen, Germany). For superresolution microscopy, images were taken with an Elyra P.1 (Zeiss) miRNA profiling.

We used the Discovery Panel from Firefly Service at Abcam to profile 408 miRNAs. Aliquots of 40 μl of 270 pg/μl of RNA (samples included vHD, vLD and CL) were shipped to the Abcam service and the profiling was done according to their standards. Samples were normalized to total amount of RNA. The data used for the analysis was raw data with background subtracted. Only data with more than 0.1 arbitrary units of fluorescence (AUF) was considered for further analysis.

## RNA extraction and TGIRT-seq library preparation

Cellular and EV RNAs were extracted by using a mirVana miRNA Isolation Kit. Cellular RNA was extracted using a modified version of the mirVana protocol, enriching for <200 nt transcripts, in order to reflect EV RNA composition by transcript size. EV RNA was extracted using the standard miRVana protocol. TGIRT-seq libraries were prepared from 2 to 10 ng of starting material.

TGIRT-seq libraries were prepared essentially as described (*Qin et al., 2016*), with a modification in the starting molecule (see below). Reverse transcription reactions contained purified RNAs, buffer

(20 mM Tris-HCl, pH7.5, 450 mM NaCl, 5 mM MgCl$_2$), 5 mM DTT, 100 nM starting molecule (see below) and 1 µM TGIRT-III (Ingex). Reverse transcription by TGIRT-III is initiated by template switching from a starting molecule consisting of a DNA primer (5'-GTGACTGGAGTTCAGACGTGTGCTC TTCCGATCTATTAN-3') encoding the reverse complement of the Illumina Read2 sequencing primer binding site (R2R) annealed to a complementary RNA oligonucleotide (R2) such that there is a single nucleotide 3' DNA overhang composed of an equimolar mixture of A, G, C and T. The RNA oligonucleotide is blocked at its 3' end with C3Sp (IDT) to inhibit template switching to itself. Reactions were pre-incubated at room temperature for 30 min and then initiated by addition of 1 mM dNTPs. Reactions were then incubated at 60°C for 15 min and terminated by adding 5 N NaOH to a final concentration of 0.25 N and incubated at 95°C for 3 min to degrade RNAs and denature protein. The reactions were then cooled to room temperature and neutralized with 5 N HCl. cDNAs were purified by using a Qiagen MinElute Reaction Cleanup Kit and then ligated at their 3' ends to a modified DNA oligonucleotide (5'Phos- CCTGTTATCCCTAGATCGTCGGACTGTAGAACTCTGAACGTG TAC-3'-C3Sp) encoding the reverse complement of the Illumina Read1 primer binding site (R1R) using Thermostable 5' AppDNA/RNA Ligase (New England Biolabs). Ligated cDNAs were re-purified with MinElute Reaction Cleanup Kit and amplified by PCR for 12 cycles using Phusion DNA polymerase (Thermo Fisher Scientific) with overlapping multiplex and barcode primers that add sequences necessary for Illumina sequencing. PCR reactions were purified using a Select-a-size DNA Clean and Concentrator (Zymo) to remove adapter dimers. Libraries were sequenced on a NextSeq 500 instrument (75-nt, single end reads) at the Genome Sequencing and Analysis Facility at the University of Texas at Austin.

## RNA sequencing analysis

Illumina TruSeq adapters and PCR primer sequences were trimmed from the reads with cutadapt 1.16 (*Martin, 2011*) (sequencing quality score cut-off at 20) and reads <15 nt after trimming were discarded. Reads were then mapped using HISAT2 v2.0.2 (*Kim et al., 2015*) with default settings to the human genome reference sequence (Ensembl GRCh38 Release 76) combined with additional contigs for 5S and 45S rRNA genes and the *E. coli* genome sequence (Genebank: NC_000913) (denoted Pass 1). The additional contigs for the 5S and 45S rRNA genes included the 2.2 kb 5S rRNA repeats from the 5S rRNA cluster on chromosome 1 (1q42, GeneBank: X12811) and the 43 kb 45S rRNA repeats that contained 5.8S, 18S and 28S rRNAs from clusters on chromosomes 13,14,15,21, and 22 (GeneBank: U13369). Unmapped reads from Pass 1 were re-mapped to Ensembl GRCh38 Release 76 by Bowtie 2 v2.2.6 (*Langmead and Salzberg, 2012*) with local alignment to improve the mapping rate for reads containing post-transcriptionally added 5' or 3' nucleotides (e. g., CCA and poly(U)), short untrimmed adapter sequences, or non-templated nucleotides added to the 3' end of the cDNAs by the TGIRT enzyme (denoted Pass 2). The mapped reads from Passes 1 and 2 were combined using Samtools 1.8 (*Li et al., 2009*) and intersected with gene annotations (Ensembl GRCh38 Release 76) supplemented with the RNY5 gene and its 10 pseudogene sequences, which were not annotated in this release, to generate the counts for individual features. Coverage of each feature was calculated by Bedtools (*Quinlan and Hall, 2010*). To avoid mis-mapping reads with embedded sncRNAs, reads were first intersected with sncRNA annotations and the remaining reads were then intersected with the annotations for protein-coding genes, lincRNAs, antisense, and other lncRNAs. To further improve the mapping rate for tRNAs and rRNAs, we combined reads mapped to tRNAs or rRNAs in the initial alignments and re-mapped them to tRNA reference sequences (Genomic tRNA Database, and UCSC genome browser website) or rRNA reference sequences (GeneBank: X12811 and U13369) using Bowtie two local alignment. Because similar or identical tRNAs with the same anticodon may be multiply mapped to different tRNA loci by Bowtie 2, mapped tRNA reads were combined according to their anticodon (N = 48) prior to calculating the tRNA distributions.

For miRNA analysis, sequences were mapped to miRbase using miRdeep2. Reads were normalized by dividing the numbers or reads per miRNA to the total number of miRNA reads in each sample and then this value was multiplied by one million (reads per million miRNA mapped reads – RPM).

## CRISPR/Cas9 genome editing

A pX330-based plasmid expressing Venus fluorescent protein (*Shurtleff et al., 2016*) was used to clone the gRNAs targeting Rab27a and Rab35. Two CRIPSR guide RNAs targeting each gene were selected using the CRISPR design tool (*Hsu et al., 2013*). For Rab27a, gRNAs targeting exon 4 and exon five were selected. For Rab35 both gRNAs targeted exon 1. Both gRNAs targeting each gene were cloned simultaneously in the pX330-Venus plasmid according to the PrecisionX Multiplex gRNA Kit instructions (SBI). MDA-MB-231 cells at 25% confluency were transfected for 48 hr, and then, they were trypsinized and sorted for single Venus-positive cells in a 96 well plate using a BD Influx cell sorter. Wells containing clones were allowed to expand (30 clones for Rab27a and 25 for Rab35) and knockouts for each gene were confirmed by immunoblots.

## CRISPR interference

MDA-MB-231 cells expressing dCas9-KRAB, as in *Gilbert et al. (2013)*, were generated using lentivirus. A modified version of the transfer plasmid, as in *Gilbert et al. (2013)*, UCOE- EF1α-dCas9-BFP-KRAB, was kindly provided by Jonathan Weissman (UCSF). Cells were bulk sorted for BFP signal 3 d post transduction and selected cells were expanded by growth for a few generations, and then frozen and stored as parental cells (these cells are referred to as WT throughout the manuscript). Sequences for gRNAs targeting the promoter of the genes of interest were extracted from *Horlbeck et al. (2016)*. gRNAs were cloned in plasmid pu6-sgRNA EF1Alpha-puro-T2A-BFP (*Gilbert et al., 2014*), plasmid #60955 obtained from Addgene, following the cloning protocol as in *Gilbert et al. (2014)*. The three top gRNAs from the V.2 library (*Horlbeck et al., 2016*) were chosen per gene of interest. Lentiviruses with the gRNAs targeting the genes of interest were use to transduce the parental cells. Three days post transduction cells were selected with 2 ug/ml puromycin for 3 d. Post puromycin selection, cells were collected for up to three generations (~72 hr) for best levels of protein depletion. More doubling times showed reduced protein depletion.

## In vitro packaging reactions

### Membrane and cytosol preparation

Fractionation of cells and membranes was done as in *Shurtleff et al. (2016)* with some modifications as indicated. MDA-MB-231 cells were harvested at ~80% confluency by adding cold PBS and physically removing the cells by the use of a cell scrapper. Cells were then centrifuged at 1000xg for 10 min at 4°C and cell pellets were frozen at −80°C until use. Cells were thawed and resuspended in 2 volumes of HB buffer (20 mM HEPES pH 7,4, 250 mM sorbitol) containing protease inhibitor cocktail l (1 mM four aminobenzamidine dihydrochloride, 1 mg/ml antipain dihydrochloride, 1 mg/ml aprotinin, 1 mg/ml leupeptin, 1 mg/ml chymostatin, 1 mM phenymethylsulfonyl fluoride, 50 mM N-tosyl-L-phenylalanine chloromethyl ketone and 1 mg/ml pepstatin). Cells were passed 21–25 times through a 22 gauge needle until >80% of cells were lysed as assessed by microscopy and trypan blue staining. All steps from hereon were done on ice and 4°C, unless otherwise specified. The homogenized cells were centrifuged at 1500xg for 20 min and the supernatant fraction was subsequently centrifuged at 15,000xg for 15 min to collect donor membranes using a FA-45-30-11 rotor and an Eppendorf 5430 centrifuge (Eppendorf, Hamburg, Germany). The supernatant fraction was centrifuged again at ~150,000 xg (49,000 RPM) in a TLA-55 rotor and Optima Max XP ultracentrifuge (Beckman Coulter) to generate the cytosol fraction (5–6 mg/ml). The 15,000xg membrane fraction (pellet) was resuspended in 1 vol of HB buffer and an equal volume of 0.8M LiCl. Donor membranes were then centrifuged again as before and the pellet fraction was resuspended in 1 vol of original starting material HB buffer.

### In vitro miRNA packaging reaction

The in vitro packaging reaction was performed as described in *Shurtleff et al. (2016)*. Briefly, miR-122–5 p and the variant versions were purchased from Integrated DNA Technologies (IDT, Coralville, IA). Membranes and cytosols were prepared from MDA-MB-231 cells as described above, with the exception of assays using YBX1 KO cells, in which case HEK293T cells WT and YBX1 KO were used as described *Shurtleff et al. (2016)*. Complete miRNA packaging assays consist of 10 ul membranes, 16 ul cytosol (the cytosolic concentration was 5.5 mg/ml, unless otherwise specified), 4 ul 10X ATP regeneration system (400 mM creatine phosphate, 2 mg/ml creatine phosphokinase, 10

mM AT, 20 mM HEPES pH 7.2, 250 mM sorbitol, 150 mM KOAc, 5 mM MgOAc), 8 ul 5X incorporation buffer (400 mM KCl, 100 mM CaCl2, 60 mM HEPES-NaOH, pH 7.4, 6 mM, MgOAc), 1 ul of 10 nM synthetic single stranded RNA and 1 ul RNAsin (Promega, Madison, WI). For rescue experiments with purified La,~0.1 ug of total purified La was added per reaction. Reactions were incubated for 20 min at 30℃, then 1 ul of RNase If (50,000units/ml) (NEB, Ipswich, MA) and 4 ul of 10X NEB buffer 3 (NEB) was added to the reactions and incubated for an extra 20 min at 30℃. Following incubation, RNA was extracted using Direct-Zol (Zymo Research, Irvine, CA) according to manufacturer's instructions and miRNA was quantified using TaqMan miRNA assays as described above. The output was represented as a percentage of protection by comparing the level of miRNA left in the RNase-treated samples relative to an ice control which was not RNase-treated. The ice, no RNase control was set to 100%.

## Streptavidin pull-down of miR-122 and interacting proteins

The in vitro packaging reaction was performed as described above with a modified version of miR-122, 3' biotinylated miR-122 (IDT), which was used to mediate its capture with streptavidin beads. Post-incubation with RNase If, the reactions were heated to 65℃ for 15 min to inactivate RNase If and then mixed with 10% TX-100 for a final concentration of 1% and kept on ice for 30 min. Dynabeads Myone Streptavidin T1 (Thermo Fischer Scientific) were washed three times with 1X incorporation buffer and then added to the reactions. The reaction was incubated for 1.5 hr at 4℃ with constant rotation. Beads were washed with 1X incorporation buffer five times and bound proteins were eluted with 1X Laemmli buffer without bromophenol blue. Proteins were electrophoresed in a 4–20% acrylamide Tris-Glycine gradient gel (Life Technologies) for ~3 min. The bulk of proteins were stained with Coomassie and the stained band was excised from the gel using a fresh razor blade. Samples were submitted to the Vincent J. Coates Proteomics/Mass Spectroscopy laboratory at UC Berkeley for in-gel tryptic digestion of proteins followed by liquid chromatography and mass spectrometry analysis according to their standards. The list of detected proteins was then curated for RNA binding proteins, excluding any structural ribosomal protein.

## Protein purification

Human La protein was expressed in Sf9 cells as a N-terminal 6x his tagged version pLJM60-Ssb (*Thoreen et al., 2012*), plasmid #38241 obtained from Addgene, served as the La sequence backbone. La was cloned in the pFastBac vector from the Bac-to-Bac Baculovirus expression System (Thermo Fischer Scientific) according to the manufacturer's specifications. The generation of bacmids and the production of baculoviruses was performed as indicated by the manufacturer. For La purification, 1 liter of Sf9 cells was infected with baculovirus. Cells expressing La were harvested 2 d post infection by centrifugation at 1000xg for 15 min at 4℃ using a Sorvall RC 6+ centrifuge with a FIBER-Lite F10−6 × 500Y rotor. Cellular pellet fractions were stored at −20℃ until use. The cellular pellet was thawed and resuspended in 40 ml of lysis buffer (20 mM HEPES, pH8, 0.1 mM EGTA, 500 mM NaCl, 5 mM DTT, 10 mM imidazole, 10 mM MgCl$_2$, 250 mM sorbitol, 5% glycerol, 1 mM PMSF, and protease inhibitor cocktail 1 mM 4-aminobenzamidine dihydrochloride, 1 µg/ml antipain dihydrochloride, 1 µg/ml aprotinin, 1 µg/ml leupeptin, 1 µg/ml chymostatin, 1 mM phenymethylsulfonyl fluoride, 50 µM N-tosyl-Lphenylalanine chloromethyl ketone and 1 µg/ml pepstatin)), homogenized and then sonicated. Post-lysis, the material was centrifuged at ~40,000 xg (20,000 RPM) using a FIBERlite F211−8 × 50Y rotor for 20 min at 4℃. The supernatant fraction was centrifuged again at ~42,500 xg for 1 hr using a Ti70 rotor. In preparation for the sample, 2 ml of Ni-NTA resin beads (Thermo Fischer Scientific) were washed twice with lysis buffer. The supernatant then was mixed with Ni-NTA beads and the suspension incubated with constant rotation at 4℃ for 2.5 hr. Beads were washed three times with washing buffer (recipe for washing buffer was the same as lysis buffer (above) but with 50 mM imidazole instead of 10 mM imidazole). Proteins were eluted with 4 ml of elution buffer (elution buffer similar recipe to lysis buffer, but the imidazole concentration was 250 mM). Eluted proteins were applied to a Sephadex G25 column (Thermo Fischer Scientific). Buffer was exchanged to Buffer B (10 mM Tris, pH 7,4, 150 mM NaCl, 3 mM MgCl2, 5% glycerol). Purified fractions were pooled, flash-frozen in liquid nitrogen and stored at −80 C.

## Electrophoretic mobility shift assays

In-gel fluorescence was detected in order to assess the free and protein-bound RNA. Fluorescently labeled (5') RNAs (IRD800CWN) were ordered from Integrated DNA Technologies (IDT, Coralville, IA). EMSAS were performed following *Rio (2014)*, with some modifications. In brief, 1 nM of fluorescently labeled RNA was mixed and incubated with increasing amounts of purified La, ranging from 250pM – 2 uM. Buffer E was used in this incubation (25 mM Tris pH8, 100 mM KCl, 1.5 mM MgCl2, 0.2 mM EGTA, 0.05% Nonidet P-40, 50 ug/ml heparin). Reactions were incubated at 30°C for 30 min and then chilled on ice for 10 min and mixed with 6X loading buffer (60 mM KCl, 10 mM Tris pH 7,6, 50% glycerol, 0.03% (w/v) xylene cyanol). Polyacrylamide gels (6%), acrylamide:bisacrylamide (29:1) (Bio-Rad, Hercules, CA), made with Tris-glycine buffer were prerun for 30 min at 200V in the cold room. Samples were resolved in the prerun gels at 200V for 50 min in a 4°C cold room. Fluorescence was detected by using an Odyssey CLx Imaging System (LI-COR Biosciences, Lincoln, NE). The software of the Odyssey CLx Imaging System was used to obtain quantification of fluorescence. To calculate $K_d$s, we fit Hill equations with quantified data points. Fraction bound was calculated as a function of exhaustion of free-miRNA.

## Immunoprecipitation of La and La-RNA complexes

Approximately 40 million cells were harvested as described in the section 'Membrane and Cytosol preparation.' Cells were homogenized in 2 volumes of HB buffer and physically disrupted as previously described. Non-lysed cells and nuclei were centrifuged at 1500xg for 20 min. The supernatant fraction was used as source of cytoplasmic La for immunoprecipitation and was divided into three equal parts: input, La-IP, beads only. Dynabead Protein G (Thermo Fischer Scientific) was washed three times in polysome lysis buffer (*Peritz et al., 2006*). A 5X polysome lysis buffer was made and mixed with 1500xg post-nuclear supernatant to a final 1X concentration. Anti-La (Origene Technologies, #TA500406) was added to the post nuclear supernatant to a final concentration of 2 ug/500 ul and the mixture was incubated with rotation overnight at 4°C. Dynabeads Protein G (Thermo Fischer) beads were added to the mixture and this mixture was incubated for an additional 3 hr at 4°C with constant rotation. Beads were washed 5 times with 1X polysome buffer and the content was divided for protein or RNA analysis. Beads for protein analysis were incubated with Laemmli buffer and heated at 95°C for 10 min and beads for RNA analysis were exposed to TRI reagent (Zymo Research) for RNA extraction using Direct-Zol (Zymo Research) as indicated by the manufacturer. Protein and miRNAs were analyzed by immunoblots and TaqMan miRNA qPCR analysis, respectively.

## Motif analysis

For motif analysis, we used the MEME Suite 5.0.2 (*Bailey and Elkan, 1994*). MiRNAs detected uniquely in the vHD sub-population by TGIRT-seq were pooled together with miRNAs that were found to be enriched by at least 10-fold (vHD/cellular lysate) through TGIRT-seq and Firefly profiling. A total of 49 miRNAs met those requirements and were used for further analysis. A 0 and 1 Markov background model was used to find 3–5 nucleotide motifs. The total mature *Homo sapiens* miRNAs from miRBase were used as background.

# Acknowledgements

We thank Kathleen Collins, Arash Komeili and Donald Rio for advice. We also thank Kathleen Collins and Daniel Sirkis for reading and editing the manuscript. We would also like to thank the staff at the UC Berkeley shared facilities, the DNA sequencing facility, the Vincent J Coates Proteomics Facility, the Flow Cytometry Facility, the Cell Culture Facility and the Genomic Sequencing and Analysis Facility at UT Austin. Cartoons were created with Biorender (Toronto, Ontario, Canada).

# Additional information

**Competing interests**

Randy Schekman: Reviewing Editor and Founding Editor-in-Chief, *eLife*. Alan M Lambowitz: Thermostable group II intron reverse transcriptase (TGIRT) enzymes and methods for their use are the

subject of patents and patent applications that have been licensed by the University of Texas and East Tennessee State University to InGex, LLC. AML, some former and present members of the AML laboratory, and the University of Texas are minority equity holders in InGex, LLC and receive royalty payments from the sale of TGIRT enzymes and kits and from the sublicensing of intellectual property to other companies. The other authors declare that no competing interests exist.

## Funding

| Funder | Grant reference number | Author |
|---|---|---|
| Howard Hughes Medical Institute | | Randy Schekman |
| National Institutes of Health | R01 GM37949 | Alan M Lambowitz |

The funders had no role in study design, data collection and interpretation, or the decision to submit the work for publication.

## Author contributions

Morayma M Temoche-Diaz, Conceptualization, Data curation, Formal analysis, Validation, Investigation, Visualization, Methodology, Writing—original draft, Writing—review and editing; Matthew J Shurtleff, Conceptualization, Formal analysis, Visualization, Methodology, Writing—review and editing; Ryan M Nottingham, Formal analysis, Investigation, Writing—review and editing; Jun Yao, Data curation, Formal analysis, Writing—review and editing; Raj P Fadadu, Investigation, Writing—review and editing; Alan M Lambowitz, Resources, Supervision, Funding acquisition, Methodology, Writing—review and editing; Randy Schekman, Conceptualization, Resources, Supervision, Funding acquisition, Methodology, Writing—review and editing

## Author ORCIDs

Morayma M Temoche-Diaz (iD) https://orcid.org/0000-0003-1119-1749
Matthew J Shurtleff (iD) http://orcid.org/0000-0001-9846-3051
Ryan M Nottingham (iD) http://orcid.org/0000-0001-6937-5394
Jun Yao (iD) https://orcid.org/0000-0002-1232-1587
Raj P Fadadu (iD) https://orcid.org/0000-0002-7163-9237
Alan M Lambowitz (iD) https://orcid.org/0000-0001-6036-2423
Randy Schekman (iD) https://orcid.org/0000-0001-8615-6409

## Decision letter and Author response

Decision letter https://doi.org/10.7554/eLife.47544.030
Author response https://doi.org/10.7554/eLife.47544.031

# Additional files

## Supplementary files

• Transparent reporting form
DOI: https://doi.org/10.7554/eLife.47544.026

## Data availability

RNA sequencing data have been deposited in SRA under accession code PRJNA532890.

The following dataset was generated:

| Author(s) | Year | Dataset title | Dataset URL | Database and Identifier |
|---|---|---|---|---|
| Temoche-Diaz MM, Shurtleff MJ, Nottingham RM, Yao J, Fadadu RP, Lambowitz AM, Schekman R | 2019 | TGIRT-seq of extracellular vesicle sub populations released from MDA-MB-231 cells | https://www.ncbi.nlm.nih.gov/sra/PRJNA532890 | Sequence Read Archive, PRJNA532890 |

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
