## [Decision Letter]

Thank you for submitting your article "Distinct mechanisms of microRNA sorting into cancer cell-derived extracellular vesicle subtypes" for consideration by *eLife*. Your article has been reviewed by two peer reviewers, one of whom is a member of our Board of Reviewing Editors, and the evaluation has been overseen by a Reviewing Editor (Suzanne Pfeffer) and Jeffrey Settleman as the Senior Editor.

The reviewers have discussed the reviews with one another and the Reviewing Editor has drafted this decision to help you prepare a revised submission. One of the reviewers has included a number of very detailed comments that should be easily addressed.

Summary:

The present study by Temoche-Diaz et al., reports that Lupus La protein packages a subset of miRNAs for export in a subset of endosome-derived extracellular vesicles (EVs). The authors used ultracentrifugation followed by iodixanol-based centrifugation to obtain two biochemically distinct "EV subtypes" (LD and HD) in which they find different, significantly enriched miRNAs. The authors further provide evidence that La binds miR-122 (specifically enriched in HD EVs) to mediate its targeting within exosomes. Using a semi-reconstituted cell-free assay and binding experiments the authors conclude that La specifically engages a subset of microRNAs in the exosome biogenesis pathway. The work will be of broad interest if the authors can determine the number of miRNAs that depend on La for their packaging. Also, a recently published paper should be discussed in relation to the current story.

Jeppesen et al., 2019 analyzed different cell types (including the MDA-MB-231 cell line) using a similar approach to isolate two distinct fractions, LD and HD. Jeppesen et al., extended their analysis by including IPs using specific tetraspanins (CD63, CD81, CD9) to further isolate vesicles from non-vesicular particles (exomeres, lipoproteins and other RNA-binding protein complexes), the latter found to be mostly enriched in the HD fraction. Their data on CD63 versus CD9 seem to differ from the present story but could this be a cell-type dependent effect? This must be answered.

Previous work by Schekman and colleagues also used anti-CD63 immunoprecipitation to isolate vesicles enriched for microRNAs. La protein can interact with major vault protein (MVP)-particles (Kickhoefer et al., 2002), an RNA-binding proteinaceous organelle that has previously been shown to be export microRNAs by poorly understood mechanisms (Jeppesen et al., 2019, Teng et al., 2017). Vault particles were enriched in Coffey's iodixanol HD fractions of MDA-MB-231 cells.

Essential revisions:

1) Jeppesen et al., clearly showed that while their "LDs" are almost exclusively enriched in vesicles, the predominant components of the HD fraction are non-vesicular structures. Thus it would be important for the authors to show that miR-122 is detected in anti-CD63 immuno-isolated vesicles from the HD fraction, and that miR-122 export is dependent on Rab27A. Figure 6C shows a sucrose flotation performed on isolated HD iodixanol fractions, which is consistent with these authors' conclusion, but anti-CD63 IP would add a lot.

2) The authors should blot for an additional non-vesicular particle marker (MVP or Ago?) for their Optiprep gradient fractions. The blots for CD63 or TSG101 are rather smeary across these fractions. Typically, CD63 runs as a smear due to glycosylation and only a few antibodies work on gels run under reducing conditions. Please provide molecular weight markers for the blots and consider whether your antibody is appropriately recognizing CD63 as it seems to have a strange pattern (2 discrete bands) and to be present across the whole blot, not really enriched in any fractions. Jeppesen et al., showed Ago1 and Ago2 in the HD fraction of MDA-MB-231 cells. Please at least discuss this parallel study in relation to the current findings.

3) Does depletion of La protein change the overall miRNA signature of the LD and HD fractions? This would provide important information as to how many miRNAs rely on the proposed La pathway for export and would add much to the significance of the present story.

4) Immunoprecipitated La in Figure 7 may co-immunoprecipitate vault proteins (MVPs), which may also be needed for export of miR-122. The authors should show in Figure 8 the non-binding miRNA sequence for comparison.

5) It is not clear that Rab27a-KO is really making the HD population go away – please show this with a density gradient analysis, with quantitation of vesicle numbers and Western blot analysis of the gradient fractions. It also seems strange that the HD fraction and not the LD fraction would go away with Rab27a KO unless the method you use here is much different from that used in Kowal et al. in which the LD fraction was thought to represent exosomes. Please clarify.

[Editors' note: further revisions were requested prior to acceptance, as described below.]

Thank you for resubmitting your work entitled "Distinct mechanisms of microRNA sorting into cancer cell-derived extracellular vesicle subtypes" for further consideration at *eLife*. Your revised article has been favorably evaluated by Jeffrey Settleman (Senior Editor) and Suzanne Pfeffer as BRE.

There is no question that the manuscript has been improved and we would like to publish the story in *eLife*. We just wanted to give you a final chance to include some of your rebuttal discussion in the text of the article so that the readers will also have a better understanding of some of the issues raised by the reviewers. We will abide by your decision on this, but believe that readers will benefit from inclusion of some of these points.

---

## [Author Response]

Summary:The present study by Temoche-Diaz et al., reports that Lupus La protein packages a subset of miRNAs for export in a subset of endosome-derived extracellular vesicles (EVs). The authors used ultracentrifugation followed by iodixanol-based centrifugation to obtain two biochemically distinct "EV subtypes" (LD and HD) in which they find different, significantly enriched miRNAs. The authors further provide evidence that La binds miR-122 (specifically enriched in HD EVs) to mediate its targeting within exosomes. Using a semi-reconstituted cell-free assay and binding experiments the authors conclude that La specifically engages a subset of microRNAs in the exosome biogenesis pathway. The work will be of broad interest if the authors can determine the number of miRNAs that depend on La for their packaging. Also, a recently published paper should be discussed in relation to the current story.Jeppesen et al., 2019 analyzed different cell types (including the MDA-MB-231 cell line) using a similar approach to isolate two distinct fractions, LD and HD. Jeppesen et al., extended their analysis by including IPs using specific tetraspanins (CD63, CD81, CD9) to further isolate vesicles from non-vesicular particles (exomeres, lipoproteins and other RNA-binding protein complexes), the latter found to be mostly enriched in the HD fraction. Their data on CD63 versus CD9 seem to differ from the present story but could this be a cell-type dependent effect? This must be answered.

Jeppesen et al., developed a method (“high-resolution density gradient”) to isolate two fractions they termed LD and HD. The two tetraspanins they focused their studies on when analyzing fractions from their high-resolution density gradient are CD63 and CD81. There is different distribution of both tetraspanins in their high-resolution density gradient for MDA-MB-231-derived sEVs as shown below: CD81 has two major peaks, one coinciding with the CD63 peak, and one showing a lighter density. The present study has not analyzed the distribution of CD81, however studies have shown that both CD81 and CD9 positive vesicles bud from the plasma membrane (Deneka et al., 2007) and that CD9 and CD81 have mainly a plasma membrane sub-cellular localization (Gould, Fordjour and Daaboul, 2019). Therefore, the CD81 distribution to lighter fractions might mirror the CD9 distribution we show in the present in our study (Figure 1B).

Here we report the purification of two sEV sub-populations based on their distinct equilibrium buoyant density distributions. The sEV sub-populations reported here have different enrichments of CD9 and CD63 proteins. Jeppesen et al. results on CD63 IPs in DKO-1 and Gli36 cell lines (no IPs for MDA-MB-231 were shown) showed that CD9 co-IPs with CD63, however the amount of CD9 co-IPing with CD63 and the amount that did not, was not shown. Therefore, it is possible that some, but not all CD9 co-IPs with CD63. Previous studies have shown that there are at least two CD9 sEV sub-populations, one that is immunoisolated with CD63 and one that is not; the later possibly having its origin in the plasma membrane (Kowal et al., 2016). In agreement with the IP results from Kowal et al., we see some CD9 positive vesicles co-fractionating with CD63 in our linear gradient, however most of our CD9 fractionates away from the CD63 positive fraction. The routes of CD63 and CD9 positive vesicles biogenesis might be cell-line specific.

Previous work by Schekman and colleagues also used anti-CD63 immunoprecipitation to isolate vesicles enriched for microRNAs. La protein can interact with major vault protein (MVP)-particles (Kickhoefer et al., 2002), an RNA-binding proteinaceous organelle that has previously been shown to be export microRNAs by poorly understood mechanisms (Jeppesen et al., 2019, Teng et al., 2017). Vault particles were enriched in Coffey's iodixanol HD fractions of MDA-MB-231 cells.Essential revisions:1) Jeppesen et al., clearly showed that while their "LDs" are almost exclusively enriched in vesicles, the predominant components of the HD fraction are non-vesicular structures. Thus it would be important for the authors to show that miR-122 is detected in anti-CD63 immuno-isolated vesicles from the HD fraction, and that miR-122 export is dependent on Rab27A. Figure 6C shows a sucrose flotation performed on isolated HD iodixanol fractions, which is consistent with these authors' conclusion, but anti-CD63 IP would add a lot.

Jeppesen et al., developed an iodixanol linear gradient to isolate two fractions they term LD and HD. Their methodology varies from ours in the sense that their iodixanol gradients span from 12-36% and they collect 1ml fractions. Ours span from 5-25% and 400µl fractions are collected. Thus, our methodology yields improved resolution and detection of different extracellular vesicle populations. On the other hand, Jeppesen et al., refer to their vesicular fractions as LD and the non-vesicular fractions as HD. Here, we further fractionate what they term LD into two distinct sEV sub-populations. We recognize the confusion and now refer to the two sEV sub-populations in this study as vesicular LD and vesicular HD to prevent confusion (vLD and vHD respectively). Jeppesen et al., presented EM images as support that their LD and HD fractions correspond to vesicular and non-vesicular fractions. In Author response image 2 we present evidence of the vesicular nature of the vLD and vHD fractions from this study by EM images.

**Author response image 2. respfig2:** 

We have also included Figure 3—figure supplement 4 to show that most of miR-122 co-fractionates with vHD, floating away from the non-vesicular associated RNP components dicer and Ago2, both of which remain in higher densities in our iodixanol gradient.

In addition to fractionation, we use RNase protection assays to further support the nature of miR-122 as sEV resident (Figure 3F). We show that extracellular miR-122 is protected from RNases unless treated with non-ionic detergent which renders it susceptible. Combined these results support our conclusion that miR-122 resides in the lumen of sEVs.

2) The authors should blot for an additional non-vesicular particle marker (MVP or Ago?) for their Optiprep gradient fractions. The blots for CD63 or TSG101 are rather smeary across these fractions. Typically, CD63 runs as a smear due to glycosylation and only a few antibodies work on gels run under reducing conditions. Please provide molecular weight markers for the blots and consider whether your antibody is appropriately recognizing CD63 as it seems to have a strange pattern (2 discrete bands) and to be present across the whole blot, not really enriched in any fractions. Jeppesen et al. showed Ago1 and Ago2 in the HD fraction of MDA-MB-231 cells. Please at least discuss this parallel study in relation to the current findings.

We have included Ago2 and dicer in Figure 1B. We are also including another previously shown iodixanol gradient (Figure 4—figure supplement 1) where Ago2 and dicer are observed on the bottom of the gradient.

The CD63 immunoblots have been replaced by new ones using the BD antibody #556019 under non-reducing conditions.

Molecular weights for all the immunoblots have been added.

3) Does depletion of La protein change the overall miRNA signature of the LD and HD fractions? This would provide important information as to how many miRNAs rely on the proposed La pathway for export and would add much to the significance of the present story.

We have included Figure 6C to address this. We tested the effect of La on the secretion of validated selectively and non-selectively sorted miRNAs by qPCR. Our data suggest that La has a broad effect on the secretion of selectively (miR-122, 142, 126, 145, 486), but not on non-selectively (miR-182, 185, 193b, 320a) sorted miRNAs.

4) Immunoprecipitated La in Figure 7 may co-immunoprecipitate vault proteins (MVPs), which may also be needed for export of miR-122. The authors should show in Figure 8 the non-binding miRNA sequence for comparison.

The miRNA that does not co-IP with endogenous La in post-nuclear supernatants from MDA-MB-231 cells is miR-182-5p. Its sequence is below:

5’ uuuggcaaugguagaacucacacu 3’

Mir-182-5p belongs to the pool of non-selectively sorted miRNAs from this study. Even though miR-182-5p contains a poly-uridine tract, it does not interact with La in our IP analysis. This could be as a result of miR-182 interaction with other RBPs or transcripts in the cytoplasm that mask it from La recognition.

5) It is not clear that Rab27a-KO is really making the HD population go away – please show this with a density gradient analysis, with quantitation of vesicle numbers and Western blot analysis of the gradient fractions. It also seems strange that the HD fraction and not the LD fraction would go away with Rab27a KO unless the method you use here is much different from that used in Kowal et al. in which the LD fraction was thought to represent exosomes. Please clarify.

Rab27a knock out only causes a decrease of CD63 positive vesicles secretion with little change in flotillin-2 (other vHD enriched marker) secretion. Therefore, Rab27a KO does not cause a depletion of the entire vHD fraction. The text and figures (Figure 2G) have been updated accordingly.

[Editors' note: further revisions were requested prior to acceptance, as described below.]

Thank you for resubmitting your work entitled "Distinct mechanisms of microRNA sorting into cancer cell-derived extracellular vesicle subtypes" for further consideration at eLife. Your revised article has been favorably evaluated by Jeffrey Settleman (Senior Editor) and Suzanne Pfeffer as BRE.There is no question that the manuscript has been improved and we would like to publish the story in eLife. We just wanted to give you a final chance to include some of your rebuttal discussion in the text of the article so that the readers will also have a better understanding of some of the issues raised by the reviewers. We will abide by your decision on this, but believe that readers will benefit from inclusion of some of these points.

We thank all the editors and the reviewers for their favorable evaluation on our resubmitted paper. We added one paragraph, subsection “Two biochemically distinct small extracellular vesicle sub-populations are released by MDA-MB-231 cells”, to discuss the results of our work in comparison to Jeppesen et al., 2019.